# SpectralGCD: Spectral Concept Selection and Cross-modal Representation Learning for Generalized Category Discovery

**Lorenzo Caselli, Marco Mistretta, Simone Magistri, Andrew D. Bagdanov**
University of Florence, Media Integration and Communication Center (MICC), Italy
`{name.surname}@unifi.it`

## Abstract

Generalized Category Discovery (GCD) aims to identify novel categories in unlabeled data while leveraging a small labeled subset of known classes. Training a parametric classifier solely on image features often leads to overfitting to old classes, and recent multimodal approaches improve performance by incorporating textual information. However, they treat modalities independently and incur high computational cost. We propose SpectralGCD, an efficient and effective multimodal approach to GCD that uses CLIP cross-modal image-concept similarities as a unified cross-modal representation. Each image is expressed as a mixture over semantic concepts from a large task-agnostic dictionary, which anchors learning to explicit semantics and reduces reliance on spurious visual cues. To maintain the semantic quality of representations learned by an efficient student, we introduce Spectral Filtering which exploits a cross-modal covariance matrix over the softmaxed similarities measured by a strong teacher model to automatically retain only relevant concepts from the dictionary. Forward and reverse knowledge distillation from the same teacher ensures that the cross-modal representations of the student remain both semantically sufficient and well-aligned. Across six benchmarks, SpectralGCD delivers accuracy comparable to or significantly superior to state-of-the-art methods at a fraction of the computational cost. The code is publicly available at: `https://github.com/miccunifi/SpectralGCD`.

## 1 Introduction

Pretrained Vision Transformers achieve remarkable performance, but fine-tuning them requires large labeled datasets. Collecting annotations is expensive, whereas unlabeled data is abundant but lacks category information. This reliance on extensive labeled datasets can limit the ability of models to adapt to unfamiliar categories in the real world. This poses challenges when encountering novel concepts, as performance is tied to the quality and breadth of the training data. *Generalized Category Discovery* (GCD) addresses this by identifying novel categories in unlabeled data while exploiting a small labeled subset of known classes to cluster unlabeled data into known (*Old*) and unknown (*New*) categories (Vaze et al., 2022; Pu et al., 2023; Rastegar et al., 2023). Unlike Novel Category Discovery, which assumes that unlabeled data contains only new categories (Han et al., 2020; Zhao & Han, 2021; Fini et al., 2021), GCD considers the general case in which unlabeled data contains both *Old* and *New* classes.

A key challenge in GCD is balancing performance on *Old* and *New* categories, as overfitting to scarce labeled data often leads models to misclassify new samples as *Old* (Liu & Han, 2025) (see Figure 1 (Left)). SimGCD addresses this via a parametric classifier on top of the image encoder (Wen et al., 2023). The classifier is trained with supervised and unsupervised cross-entropy losses plus contrastive losses on the image features. This parametric framework has been widely adopted in recent GCD approaches (Wang et al., 2024; Zheng et al., 2025; Wang et al., 2025a).

Recently, TextGCD (Zheng et al., 2025) showed that incorporating multimodal textual information via CLIP (Radford et al., 2021) significantly improves performance over unimodal approaches. To bypass the need for unknown class names, TextGCD generates LLM-based descriptions for each entry of a large lexicon, uses a frozen teacher for image-caption assignment, and then trains both image and text encoders with parametric, modality-specific classifiers.

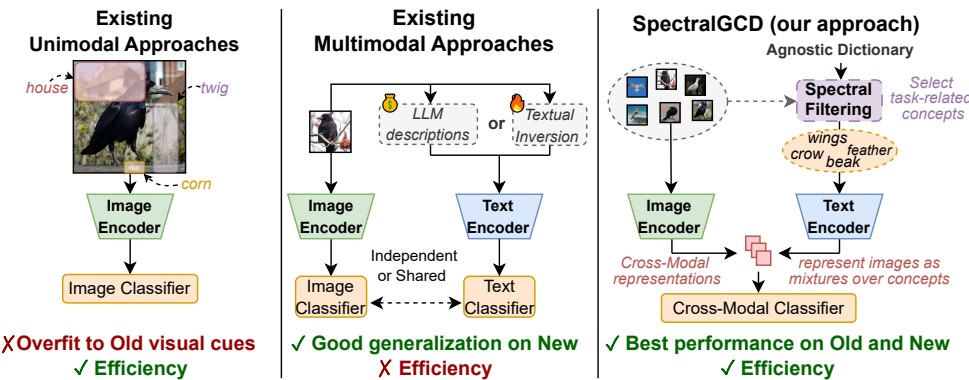

Figure 1: **Motivation and overview. (Left)** Unimodal methods are efficient, but tend to overfit to spurious visual cues. **(Center)** Introducing textual supervision improves generalization, but increases computational overhead. **(Right)** SpectralGCD leverages an agnostic dictionary by selecting task-relevant concepts, improving generalization while remaining as efficient as unimodal techniques. While existing multimodal approaches treat modalities independently, our method uses a unified, cross-modal representation to train the classifier.

More recently, GET (Wang et al., 2025a) trains a textual inversion network for pseudo-caption assignment, training the CLIP image encoder and a parametric classifier, treating each modality independently.

Although effective, these methods significantly increase training cost compared to unimodal approaches. Efficiency is crucial in GCD, as realistic deployments require periodically re-running discovery when new unlabeled data arrive (Zhao & Mac Aodha, 2023; Rypeść et al., 2025). Moreover, existing multimodal approaches treat visual and textual modalities as independent inputs to separate or shared classifier(s) (see Figure 1 (Center)). Such approaches fail to exploit the rich cross-modal relationships that vision-language models like CLIP inherently capture (Mistretta et al., 2025a).

We propose *SpectralGCD*, a multimodal GCD method inspired by probabilistic topic models (Blei et al., 2003) in which documents are represented as mixtures over latent topics. Analogously, SpectralGCD represents *images as mixtures over semantic concepts* by using CLIP's cross-modal image-text similarity scores as a unified representation. For each image, we compute cosine similarities with a large dictionary of concepts, yielding a representation that captures how strongly each concept relates to the image (e.g., a sparrow scoring high on "bird" and "wings," but low on "car" or "building"; see Figure 1 (Right)). Training a classifier directly on these cross-modal representations anchors learning to explicit semantics and reduces overfitting to spurious visual cues (Peng et al., 2025). To retain only task-relevant concepts and eliminate the need for manual annotation or noisy LLM descriptions, we introduce *Spectral Filtering* which exploits an eigendecomposition of a cross-modal covariance matrix, derived from the softmax of a strong teacher cross-modal representations. To ensure semantic quality of the representations learned by a student we employ knowledge distillation from the same frozen teacher. SpectralGCD offers an efficient approach leveraging explicit semantic concepts to guide representation learning, achieving state-of-the-art results on six benchmarks while requiring less computation than existing multimodal methods and comparable to that required by unimodal ones.

## 2 RELATED WORK

**Generalized Category Discovery (GCD).** GCD clusters unlabeled datasets containing both known and unknown categories using knowledge from a labeled set of known classes. Vaze et al. (2022) introduces the task using supervised and unsupervised contrastive learning with semi-supervised k-means on pretrained DINO representations. SimGCD (Wen et al., 2023) adds a parametric classifier and optimizes it with cross-entropy and self-distillation losses. PromptCAL (Zhang et al., 2023) uses learnable visual prompts, while SelEx (Rastegar et al., 2025) employs hierarchical semi-supervised k-means for fine-grained datasets. Recent approaches enhance visual feature representations in GCD

by increasing intra-class visual diversity, either by regularizing visual features, as in MTMC (Tang et al., 2025a), or by decomposing images into visual primitives, as in ConGCD (Tang et al., 2025b).

**Multimodal Generalized Category Discovery (Multimodal GCD).** Most existing GCD methods focus on visual information with DINO backbones. CLIP-GCD (Ouldnoughi et al., 2023) first uses CLIP (Radford et al., 2021) and textual information by concatenating visual features with text descriptions from a text corpora to improve clustering performance. More recently, GET (Wang et al., 2025a) and TextGCD (Zheng et al., 2025) better leverage CLIP and textual features. GET trains an inversion network to convert the image features to textual tokens that can be used to extract text features, and then trains a classifier, treating each modality independently. TextGCD builds a visual lexicon with LLM-generated descriptions, uses a frozen teacher for caption assignment, and trains modality-specific classifiers with soft-voting for final predictions.

In contrast, we train a parametric classifier directly on CLIP cross-modal representations (image-text cosine similarities over a concept dictionary), representing images as mixtures over semantic concepts. Unlike prior methods, we neither treat visual and textual features as independent inputs nor require additional components such as inversion networks or noisy LLM-generated text corpora.

## 3    GCD: PRELIMINARIES AND PARAMETRIC LOSSES

**Generalized Category Discovery.** GCD operates on a training dataset $\mathcal{D}$ comprising both a labeled and an unlabeled subset. The labeled subset $\mathcal{D}_l = \{(x_i^l, y_i^l)\}_{i=1}^{N_l}$ contains $N_l$ image-label pairs where each image $x_i^l \in \mathcal{X}$ is associated with a known class label $y_i^l \in \mathcal{Y}_l$. The unlabeled subset $\mathcal{D}_u = \{x_i^u\}_{i=1}^{N_u}$ contains $N_u$ images with hidden labels $\mathcal{Y}_u$, where $\mathcal{Y}_l \subset \mathcal{Y}_u$. Following standard terminology (Vaze et al., 2022; Wen et al., 2023), we denote known classes $\mathcal{Y}_l$ as *Old* and novel classes $\mathcal{Y}_u \setminus \mathcal{Y}_l$ as *New*. The goal of GCD is to cluster all unlabeled images $\mathcal{D}_u$ into $K = |\mathcal{Y}_l \cup \mathcal{Y}_u|$ categories using supervision from $\mathcal{D}_l$ and knowledge of *Old* class names only.

**Contrastive and Parametric Classification Losses.** Many unimodal and multimodal GCD methods (Wang et al., 2024; Zheng et al., 2025; Vaze et al., 2023; Lin et al., 2024; Wang et al., 2025a; Liu & Han, 2025), ours included, adopt the training framework introduced in SimGCD (Wen et al., 2023), combining contrastive learning with parametric classification losses. Given a feature extractor $f_\theta$ and a projector $\mathcal{M}$, typically a 3-layers MLP, this approach involves multiple objectives.

The *supervised* contrastive loss $\mathcal{L}_c^s$ is applied to labeled samples from the same class to encourage similar representations, while the *unsupervised* contrastive loss $\mathcal{L}_c^u$ applies to all samples, treating augmented views as positives:

$$\mathcal{L}_c^s(w_i) = \frac{1}{|\mathcal{B}^l|} \sum_{i \in \mathcal{B}^l} -\frac{1}{|P(i)|} \sum_{p \in P(i)} \log \frac{e^{(\frac{w_i^\top w_p}{\tau})}}{\sum_{j \neq i} e^{(\frac{w_i^\top w_j}{\tau})}}, \quad \mathcal{L}_c^u(w_i) = \frac{1}{|\mathcal{B}|} \sum_{i \in \mathcal{B}} -\log \frac{e^{(\frac{w_i^\top w_i'}{\tau})}}{\sum_{j \neq i} e^{(\frac{w_i^\top w_j}{\tau})}},$$

(1)

where $\mathcal{B}^l$ denotes labeled samples in the minibatch, $P(i)$ represents positive samples sharing the same label as sample $i$, $\tau$ is the temperature parameter, $w_i = \mathcal{M}(f_\theta(x_i))$ is the projected representation for the sample $x_i$ obtained by the image encoder with parameters $\theta$, and $w_i'$ is the representation of an augmented view of sample $x_i$.

A parametric classifier $L_\psi$ predicts class probabilities $p_i = L_\psi(f_\theta(x_i))$. The supervised classification loss $\mathcal{L}_{cls}^s$ uses standard cross-entropy on labeled data, while a self-distillation objective $\mathcal{L}_{cls}^u$ encourages consistency between augmented views while maximizing prediction diversity:

$$\mathcal{L}_{cls}^s(p_i, y_i) = \frac{1}{|\mathcal{B}^l|} \sum_{i \in \mathcal{B}^l} \ell(y_i, p_i), \quad \mathcal{L}_{cls}^u(p_i) = \frac{1}{|\mathcal{B}|} \sum_{i \in \mathcal{B}} \ell(p_i, p_i') - \epsilon H(\bar{p}),$$

(2)

where $\ell$ is the cross-entropy loss, $y_i$ is the ground-truth, and $p_i'$ is the predicted probability of an augmented view $x_i'$ with a lower temperature. $H(\bar{p})$ is a mean-entropy maximization term used to regularize the unsupervised self-distillation, to use the full classifier and preventing from considering only *Old* categories. It is defined as: $H(\bar{p}) = -\sum_k \bar{p}^k \log \bar{p}^k$ where $\bar{p} = \frac{1}{2|\mathcal{B}|} \sum_{i \in \mathcal{B}} (p_i + p_i')$ and $p_i$ and $p_i'$ are the probabilities of $x_i$ and $x_i'$, both with the same temperature. $\epsilon$ is a hyperparameter that weights the impact of the mean-entropy regularizer.

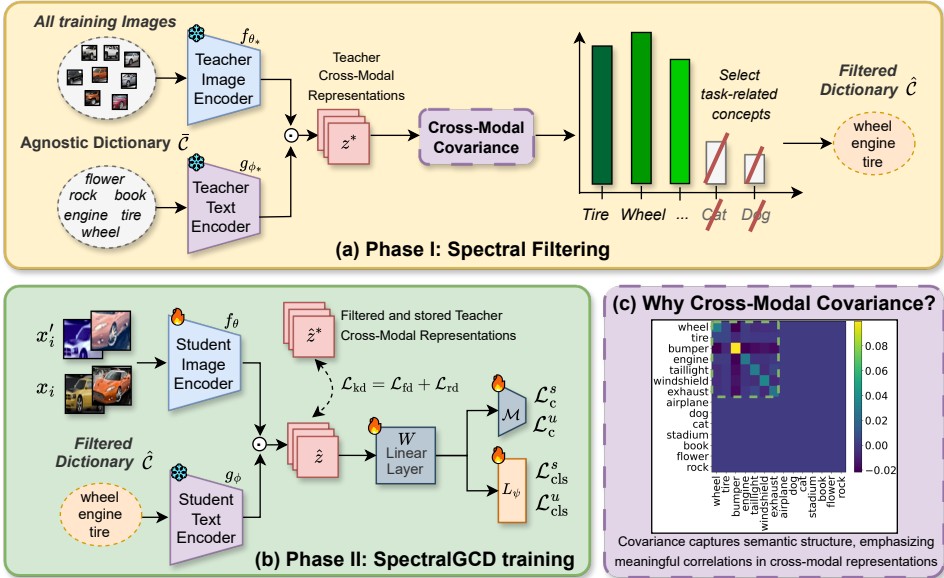

Figure 2: **The SpectralGCD two-phase approach.** **(a)** Spectral Filtering uses the cross-modal covariance computed from teacher cross-modal representations to retain only its most informative components and isolate semantically relevant concepts. **(b)** During training, we jointly optimize the image encoder $f_\theta$, linear projection $W$, classifier $L_\psi$, and MLP $\mathcal{M}$ using both parametric and contrastive objectives, while refining the semantics of our unified representation via distillation of teacher cross-modal representations computed on the filtered dictionary. **(c)** The cross-modal covariance makes explicit which concept co-activations carry meaningful signal and should be preserved.

# 4 CROSS-MODAL REPRESENTATION LEARNING VIA SPECTRALGCD

In this section we present **SpectralGCD**, a multimodal GCD approach that trains a parametric classifier directly on CLIP's image–text similarities rather than separate visual and textual streams. We motivate this with the notion of a *sufficient representation* that preserves all class-relevant information in an image and mitigates overfitting to *Old* classes. CLIP's similarities can be regarded as an approximate sufficient representation in concept space, which we call the *cross-modal representation* (Section 4.1).

Building on this, our approach proceeds in two phases: (i) **Spectral Filtering** automatically selects task-relevant concepts from a large agnostic dictionary to improve the representation (Section 4.2) and (ii) **SpectralGCD training**, where *forward and reverse distillation* from a frozen teacher preserve the semantic meaning of the cross-modal representation, while efficiently refining it during contrastive and parametric training (Section 4.3). Figure 2 gives an overview of the full pipeline.

## 4.1 TRAINING WITH CROSS-MODAL SUFFICIENT REPRESENTATIONS IN GCD

Let $x \in \mathcal{X}$ be of class $y \in \mathcal{Y}_l \cup \mathcal{Y}_u$. Suppose we have an ideal infinite text corpus of concepts $\mathcal{C}$ and an ideal mapping

$$z(\cdot; \mathcal{C}) : \mathcal{X} \to \mathbb{R}^{|\mathcal{C}|} \tag{3}$$

that assigns to each image $x$ a mixture over semantic concepts $z(x; \mathcal{C}) = [z_c]_{c \in \mathcal{C}}$, where $z_c$ measures the relevance of concept $c$ to $x$. Inspired by probabilistic topic models (Blei et al., 2003), which represent *documents as probability distributions over topics*, we represent *images as mixture of semantic concepts*. Just as a document about "machine learning" would have high topic weights for "model" and "likelihood", an image of a "sparrow" would have high $z_c$ values for concepts like "bird" and "wings", and low values for unrelated ones like "car".

If class identity $y$ depends only on such concepts, i.e. $p(y|x) = p(y|z(x; \mathcal{C}))$, then $z(x; \mathcal{C})$ is a *sufficient representation* (Achille & Soatto, 2018), meaning it captures all the information in $x$ that is relevant for classification. In this case, the Bayes-optimal classifier (Tong & Koller, 2000) can be defined entirely on $z(x; \mathcal{C})$, without needing the raw image.

Sufficient representations are crucial in GCD whose training objective combines supervised and unsupervised components (Eqs. (1) and (2)) prone to overfitting. In unimodal GCD, scarce labeled data often drives the supervised branch to exploit irrelevant visual clues, such as backgrounds, boosting performance on *Old* classes (Peng et al., 2025; Liu & Han, 2025), while the unsupervised one suffers from noisy pseudo-labels and spurious correlations (Cao et al., 2024; Zhang et al., 2025), degrading performance on New classes. Although raw image features may form approximate sufficient representations with a strong encoder, in practice they are more prone to overfitting. By representing images through semantic concepts, we aim to approximate the sufficient representation in Eq. (3), retaining class-relevant information more robustly in practice and helping to mitigate both issues.

**Parametric Classifier Training on CLIP representations.** In practice, we *empirically approximate* $z(x; \mathcal{C})$ using a large finite concept dictionary $\overline{\mathcal{C}} = \{c_j\}_{j=1}^M$ and a CLIP model with image encoder $f_\theta$ and text encoder $g_\phi$. For each image $x_i$ and concept $c_j$, we compute their cosine similarity:

$$z_{\theta,\phi}(x_i; \overline{\mathcal{C}}) = \left[ \frac{f_\theta(x_i)^\top g_\phi(c_j)}{\|f_\theta(x_i)\| \, \|g_\phi(c_j)\|} \cdot \frac{1}{\tau} \,\middle|\, c_j \in \overline{\mathcal{C}} \right] \in \mathbb{R}^M, \tag{4}$$

where $\tau$ is the CLIP's logit temperature. This yields our *cross-modal representation* $z_{\theta,\phi}(x_i; \overline{\mathcal{C}})$, a feature vector in which each entry reflects how well concept $c_j$ describes image $x_i$, similar to how Concept Bottleneck Models (Koh et al., 2020) project inputs onto interpretable concept activations. It is *cross-modal* as it combines visual and text modalities, and a *representation* since we train a parametric classifier on it, unlike standard CLIP which uses cosine similarities only for zero-shot prediction.

This representation is then projected through a linear layer $W$, which combines concept similarities into a compact embedding $u_i = W^\top z_{\theta,\phi}(x_i; \overline{\mathcal{C}})$. A parametric classifier $L_\psi$ then maps $u_i$ to class probabilities $p_i = L_\psi(u_i)$. Training uses the classification and contrastive losses from Section 3:

$$\mathcal{L}_{\text{cls}} = \lambda \mathcal{L}_{\text{cls}}^s(p_i, y) + (1-\lambda)\mathcal{L}_{\text{cls}}^u(p_i, p_i'), \qquad \mathcal{L}_{\text{c}} = \lambda \mathcal{L}_{\text{c}}^s(w_i) + (1-\lambda)\mathcal{L}_{\text{c}}^u(w_i), \tag{5}$$

where the classification branch acts on $p_i$, while the contrastive branch operates on $w_i = \mathcal{M}(u_i)$.

All losses operate on the cross-modal representation $z_{\theta,\phi}(x_i; \overline{\mathcal{C}})$: it is projected to $u_i$, mapped to class probabilities $p_i$, and to $w_i$ for contrastive learning. We train $W$ and $\psi$, together with the contrastive MLP $\mathcal{M}$, while fine-tuning only the last transformer block of $f_\theta$ (i.e. a CLIP ViT-B/16) and keeping the text encoder $g_\phi$ fixed. Training on the approximate representation $z_{\theta,\phi}(x_i; \overline{\mathcal{C}})$ anchors features to semantic concepts, reducing overfitting and spurious cues from *Old* classes compared to unimodal training (see Figure 3). This in turn enables the classifier to generalize better to *New* classes (for additional results see Appendix B.2).

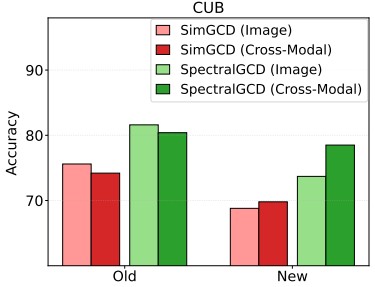

Figure 3: Comparison of SimGCD (CLIP backbone) and SpectralGCD when using image features or cross-modal representations to train the classifier. Image features are slightly better on *Old*, while cross-modal ones improve on *New*.

However, the quality of the representation $z_{\theta,\phi}(x_i; \overline{\mathcal{C}})$ depends on three factors affecting the performance. First, the dictionary $\overline{\mathcal{C}}$ is finite and may miss relevant concepts needed to describe the images; building an *agnostic* dictionary from a large text corpus improves coverage but also introduces many irrelevant concepts, adding noise to the representations. Second, even with a good dictionary, weak encoders $f_\theta$ and $g_\phi$ fail to assign meaningful weights to concepts, limiting representation quality. Finally, as the image encoder is jointly trained with the parametric classifier, the learned representation $z_{\theta,\phi}$ can drift away from semantically interpretable concepts.

To address these limitations we leverage a stronger frozen teacher CLIP, with encoders $f_{\theta_*}$ and $g_{\phi_*}$, to enhance $z_{\theta,\phi}(x_i; \overline{\mathcal{C}})$. We introduce two complementary strategies: **Spectral filtering**, which uses the teacher to retain relevant concepts and align the dictionary more closely with concepts describing the images; and **Forward and Reverse Knowledge distillation**, which transfers structural information from the teacher to preserve and refine the semantics of the cross-modal representation.

## 4.2 Concept Selection via Spectral Filtering

Our **Spectral Filtering** phase automatically selects relevant *concepts* from large agnostic dictionaries $\overline{\mathcal{C}} = \{c_j\}_{j=1}^M$ (Figure 2 (a)). This approach filters out noisy concepts in the dictionary based on the visual information contained in the entire dataset.

**Extracting Cross-Modal Covariance.** Given the teacher model with image and text encoders $f_{\theta_*}$ and $g_{\phi_*}$ respectively, we compute the cross-modal representation $z_{\theta_*,\phi_*}(x_i; \overline{\mathcal{C}})$, on the images $x_i$ and the large, task-agnostic dictionary $\overline{\mathcal{C}}$, as defined in Eq. (4).

Then, after normalization with softmax $q_i = \sigma\left(z_{\theta_*,\phi_*}(x_i; \overline{\mathcal{C}})\right) \in \mathbb{R}^M$, we compute the sample *cross-modal covariance* matrix:

$$G = \frac{1}{N-1} \sum_{i=1}^N \left(q_i - \mu\right)\left(q_i - \mu\right)^\top \in \mathbb{R}^{M \times M}, \tag{6}$$

where $N = N_l + N_u$ is the number of samples in the dataset consisting of $N_l$ labeled samples and $N_u$ unlabeled samples and $\mu$ is the empirical mean of $q_i$ across the dataset.

The eigendecomposition of $G$ yields eigenvalues $\Lambda = \{\lambda_i\}_{i=1}^M$ and eigenvectors $V = \{v_i\}_{i=1}^M$, ordered as $\lambda_1 \geq \cdots \geq \lambda_M$. Since $G$ encodes concept similarities, the eigenvalues quantify the strength of co-variation among concept activations in the cross-modal representation: large ones capture informative correlations, while small ones reflect noise. The corresponding eigenvectors distribute this variability across concepts, highlighting the most relevant ones (see Figure 2(c) for an example). We then select a subset of concepts from the task-agnostic dictionary $\overline{\mathcal{C}}$ in two stages:

**Noise Filtering.** To retain only the most informative combinations of concepts, we compute the *cumulative explained variance ratio* from the eigenvalues:

$$r_k = \frac{\sum_{i=1}^k \lambda_i}{\sum_{i=1}^M \lambda_i}. \quad k = 1, \ldots, M \tag{7}$$

Given a threshold $\beta_e \in (0,1)$, we determine $k^* < M$ as the minimum index $k$ such that $r_k \geq \beta_e$, where $k^*$ corresponds to the number of retained principal components. The resulting reduced eigenspace is determined by $\Lambda_* = (\lambda_1, \ldots, \lambda_{k^*})$ and $V_* = [v_1, \ldots, v_{k^*}]$.

**Concept Importance Selection.** From this eigenspace we compute what we call the *concept importance vector $s$*:

$$s = \sum_{i=1}^{k^*} \lambda_i \, v_i^2 \in \mathbb{R}^M, \tag{8}$$

where $v_i^2$ denotes the element-wise square of eigenvector $v_i$, and each component $s_j$ quantifies the importance of concept $c_j \in \overline{\mathcal{C}}$. We sort the components of the vector $s$ in decreasing order, $s_1 \geq \ldots \geq s_M$, obtaining a vector $\hat{s} \in \mathbb{R}^M$ associated with the concepts, and we sort the dictionary $\overline{\mathcal{C}}$ accordingly. Finally, we retain only the concepts with cumulative importance above a threshold $\beta_c \in (0,1)$ to obtain the filtered dictionary:

$$\hat{\mathcal{C}} = \{c_j \in \overline{\mathcal{C}} \mid j \leq j^*\}, \quad j^* = \min\left\{j \; \middle| \; \frac{\sum_{i=1}^j \hat{s}_i}{\sum_{i=1}^M \hat{s}_i} \geq \beta_c\right\}, \tag{9}$$

representing the relevant concepts from the large, task-agnostic dictionary $\overline{\mathcal{C}}$.

**Discriminativeness of Selected Concepts.** Spectral filtering inherently favors concepts that are discriminative for the classes in the dataset. This is because the covariance matrix is computed after a softmax normalization on teacher cross-modal representation (see Eq. (6)), which amplifies high similarity (foreground) concepts while suppressing common or weakly informative background ones. Together, with CLIP's intrinsic object bias (Schrodi et al.; Gurung et al., 2025; Dufumier et al.), this causes the dominant eigenvectors of the covariance matrix $G$ to concentrate on task-relevant object semantics (see Appendix I for an empirical analysis). The effect of softmax is qualitatively analogous to term-weighting in Latent Semantic Analysis (LSA) (Deerwester et al., 1990), a classical technique in topic modeling: both mechanisms increase the influence of salient components by redistributing the variance toward them while downplaying non-informative ones.

### 4.3 SPECTRALGCD TRAINING WITH FORWARD AND REVERSE DISTILLATION

Spectral Filtering selects relevant concepts with the *frozen teacher*, which are then used to build the cross-modal representation $z_{\theta,\phi}(x_i, \hat{\mathcal{C}})$, arising from the trainable image encoder $f_\theta$ and frozen text encoder $g_\phi$ (the *student* model). The quality of this representation depends on how effectively the student represents the teacher-selected concepts $\hat{\mathcal{C}}$. As training proceeds, however, the cross-modal student representation loses semantic meaning due to the joint training of the image encoder with the classifier (see Eq. (5)). To counter this, we apply both *forward* and *reverse* knowledge distillation, leveraging teacher guidance for cross-modal representation learning (Figure 2(b)).

We define $\hat{z}_i = z_{\theta,\phi}(x_i, \hat{\mathcal{C}})$ and $\hat{z}_i^* = z_{\theta_*,\phi_*}(x_i, \hat{\mathcal{C}})$, as the cross-modal representations of the student and the teacher, respectively, on image $x_i$ using the filtered dictionary $\hat{\mathcal{C}}$. We then apply forward and reverse distillation, yielding the overall loss:

$$\mathcal{L}_{\text{kd}} = \mathcal{L}_{\text{fd}} + \mathcal{L}_{\text{rd}}, \text{ where } \mathcal{L}_{\text{fd}} = -\frac{1}{|\mathcal{B}|}\sum_{i \in \mathcal{B}} \sigma(\hat{z}_i^*) \log \sigma(\hat{z}_i), \ \mathcal{L}_{\text{rd}} = -\frac{1}{|\mathcal{B}|}\sum_{i \in \mathcal{B}} \sigma(\hat{z}_i) \log \sigma(\hat{z}_i^*), \quad (10)$$

and $\mathcal{B}$ is a minibatch containing both labeled and unlabeled data. The forward term $\mathcal{L}_{\text{fd}}$ encourages the student to match the teacher distribution, while the reverse term $\mathcal{L}_{\text{rd}}$ sharpens student predictions by penalizing probability mass on concepts the teacher deems unlikely (Wang et al., 2025b; Mistretta et al., 2025b). The two combined terms yield a tighter alignment between student and teacher. Aligning student cross-modal representation with that of the teacher leads to improved performance (see Table 2). Moreover, these losses are *computationally efficient*: since the teacher is frozen, its representations can be precomputed once, avoiding redundant forward passes during training.

In conclusion, the combination of the distillation losses with the parametric training losses introduced in Section 4 results in the SpectralGCD training objective:

$$\mathcal{L} = \mathcal{L}_{\text{cls}} + \mathcal{L}_{\text{c}} + \mathcal{L}_{\text{kd}}. \quad (11)$$

By leveraging CLIP's cross-modal representations, our method yields a tractable representation that can be effectively and efficiently employed within parametric training losses for GCD.

## 5 EXPERIMENTAL RESULTS

In this section we report on a range of experiments comparing SpectralGCD with the state-of-the-art.

### 5.1 EXPERIMENTAL SETUP

**Datasets.** Consistent with earlier approaches (Vaze et al., 2022; Wen et al., 2023; Wang et al., 2025a; Zheng et al., 2025), we evaluate on coarse-grained classification datasets CIFAR 10/100 (Krizhevsky et al., 2009), and ImageNet-100 (Deng et al., 2009), as well as fine-grained datasets from the Semantic Shift Benchmark (Vaze et al., 2021), CUB (Wah et al., 2011), Stanford Cars (Krause et al., 2013), and FGVC-Aircraft (Maji et al., 2013) (see Appendix A for additional details).

**Implementation Details.** Following previous Multimodal GCD settings, we train only the last vision transformer block of a CLIP ViT-B/16. For spectral filtering and knowledge distillation, we use the CLIP ViT-H/14 model as the teacher[1], which is consistent with TextGCD to ensure fair comparison. When not mentioned otherwise, we use the *Tags* concept dictionary from TextGCD as default textual information source $\bar{\mathcal{C}}$. Note that TextGCD uses both the *Tags* and an LLM-based *Attributes* dictionary, rather than relying solely on *Tags*. See Table 3 for an analysis on the robustness of our approach when selecting different dictionaries. We use clustering accuracy to evaluate each method on *Old* categories, *New* categories, and their union (*All*), only on the unlabeled data following the protocol in Vaze et al. (2022) and Wen et al. (2023). All reported results are averaged over three seeds (see Appendix A for additional details).

---

[1] https://huggingface.co/laion/CLIP-ViT-H-14-laion2B-s32B-b79K

Table 1: Comparison on fine- and coarse-grained datasets with state-of-the-art Unimodal and Multimodal approaches. Results are measured in accuracy (%) for *All*, *Old*, and *New* classes. Best results are in **bold** and second best are underlined. We also report as references the zero-shot (ZS) accuracies of the student and teacher CLIP models supposing access to true class names.

| | Method | Textual Source | CUB | | | Stanford Cars | | | Aircraft | | | CIFAR-10 | | | CIFAR-100 | | | ImageNet-100 | | |
|---|---|---|---|---|---|---|---|---|---|---|---|---|---|---|---|---|---|---|---|---|
| | | | All | Old | New | All | Old | New | All | Old | New | All | Old | New | All | Old | New | All | Old | New |
| | ZS CLIP B/16 | Class Names | 56.2 | 56.1 | 56.3 | 61.3 | 67.6 | 58.2 | 28.2 | 20.8 | 31.9 | 89.3 | 88.0 | 90.0 | 68.3 | 67.3 | 70.3 | 84.0 | 84.9 | 83.5 |
| Unimodal | SimGCD | | 60.3 | 65.6 | 57.7 | 53.8 | 71.9 | 45.0 | 54.2 | 59.1 | 51.8 | 97.1 | 95.1 | 98.1 | 80.1 | 81.2 | 77.8 | 83.0 | 93.1 | 77.9 |
| | PromptCAL | ✗ | 62.9 | 64.4 | 62.1 | 50.2 | 70.1 | 40.6 | 52.2 | 52.2 | 52.3 | 97.9 | 96.6 | 98.5 | 81.2 | 84.2 | 75.3 | 83.1 | 92.7 | 78.3 |
| | SPTNet | | 65.8 | 68.8 | 65.1 | 59.0 | 79.2 | 49.3 | 59.3 | 61.8 | 58.1 | 97.3 | 95.0 | 98.6 | 81.3 | 84.3 | 75.6 | 85.4 | 93.2 | 81.4 |
| | SelEx | | 73.6 | 75.3 | 72.8 | 58.5 | 75.6 | 50.3 | 57.1 | 64.7 | 53.3 | 95.9 | 98.1 | 94.8 | 82.3 | 85.3 | 76.3 | 83.1 | 93.6 | 77.8 |
| | DebGCD | | 66.3 | 71.8 | 63.5 | 65.3 | 81.6 | 57.4 | 61.7 | 63.9 | 60.6 | 97.2 | 94.8 | 98.4 | 83.0 | 84.6 | 79.9 | 85.9 | 94.3 | 81.6 |
| Multimodal | ClipGCD | CC-12M | 62.8 | 77.1 | 55.7 | 70.6 | 88.2 | 62.2 | 50.0 | 56.6 | 46.5 | 96.6 | 97.2 | 96.4 | 85.2 | 85.0 | **85.6** | 84.0 | 95.5 | 78.2 |
| | GET | InversionNet | 77.0 | 78.1 | 76.4 | 78.5 | 86.8 | 74.5 | 58.9 | 59.6 | 58.5 | 97.2 | 94.6 | 98.5 | 82.1 | 85.5 | 75.5 | 91.7 | 95.7 | 89.7 |
| | TextGCD | Tags + Attributes | 76.6 | **80.6** | 74.7 | 86.9 | 87.4 | 86.7 | 50.8 | 44.9 | 53.8 | 98.2 | 98.0 | 98.6 | 85.7 | 86.3 | 84.6 | 88.0 | 92.4 | 85.2 |
| | TextGCD* | Tags | 69.9 | 74.9 | 67.3 | 86.2 | 91.2 | 83.8 | 48.1 | 46.3 | 49.0 | 98.3 | 97.8 | 98.5 | 83.7 | 84.7 | 81.6 | 86.6 | 92.6 | 83.6 |
| | SpectralGCD | Tags | **79.2** | 80.4 | **78.5** | **89.1** | 92.6 | **87.4** | **63.0** | 66.1 | 61.4 | **98.5** | 96.7 | **99.4** | **86.1** | **87.2** | 83.9 | **93.4** | **96.1** | **92.0** |
| | ZS CLIP H/14 | Class Names | 80.6 | 81.7 | 80.1 | 91.1 | 93.9 | 89.8 | 43.2 | 39.1 | 45.2 | 96.7 | 96.1 | 96.9 | 83.5 | 82.7 | 85.0 | 86.8 | 87.6 | 86.4 |

## 5.2 COMPARATIVE PERFORMANCE EVALUATION

**Comparison with the State-of-the-art.** Table 1 gives our comparative evaluation. We compare with a range of unimodal approaches (SimGCD (Wen et al., 2023), PromptCAL (Zhang et al., 2023), SPTNet (Wang et al., 2024), SelEx (Rastegar et al., 2025) and DebGCD (Liu & Han, 2025)), all pretrained with DINO, using the same image encoder as CLIP for fairness. For multimodal approaches, we compare ClipGCD (Ouldnoughi et al., 2023) (which uses CC-12M (Changpinyo et al., 2021) as textual source), GET (Wang et al., 2025a), TextGCD (Zheng et al., 2025), as well TextGCD using only the *Tags* dictionary as textual input, reported in the original work and denoted as TextGCD*. For TextGCD*, we additionally reproduced results on datasets not covered in the original paper.

As seen in Table 1, SpectralGCD consistently advances the state-of-the-art on both fine- and coarse-grained benchmarks. It improves *All* accuracy over TextGCD on CUB and Stanford Cars (+2.6 on CUB and +2.2 on Stanford Cars in absolute performance gain), sets a new best on FGVC-Aircraft (+1.3 compared to DebGCD), and surpasses GET on ImageNet-100 (+1.7), while achieving competitive results on CIFAR10/100.

By comparing multimodal and unimodal methods, we show that our approach – leveraging CLIP cross-modal representations – not only improves overall performance, but, like GET and TextGCD, also overcomes the overfitting of unimodal methods to *Old* classes. This is evident from the larger gap between *Old* and *New* classes in unimodal methods compared to multimodal ones (see Appendix B for additional evidence).

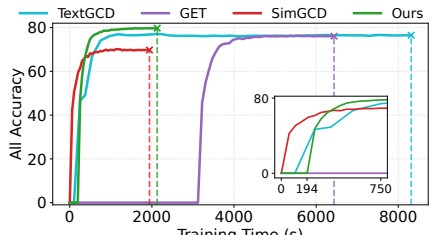

Figure 4: Accuracy vs. training time (s) for all methods on CUB. GET, TextGCD, and our method, require a preparation phase, whereas the unimodal approach SimGCD does not.

In Table 1 we also include the zero-shot performance (ZS) of the teacher (ViT H/14) and the student (ViT B/16) CLIP, by considering access to all class names. Contrary to expectations, SpectralGCD outperforms its zero-shot teacher on multiple benchmarks (e.g. +6.6 points on ImageNet-100), demonstrating that our method yields a small student model that generalizes better than the teacher.

**Training Efficiency.** In Figure 4 we report the total training times for SimGCD, GET and TextGCD on the CUB dataset, each trained for the same fixed number of epochs as in their original implementations. All methods are trained on a single RTX 4090 GPU using the original source code. State-of-the-art multimodal approaches, included SpectralGCD, require a preparation phase: GET trains an inversion network (3121 seconds), TextGCD performs image-to-textual-descriptions assignment (102 seconds), and SpectralGCD employs Spectral Filtering to select task-relevant concepts from an large agnostic dictionary (194 seconds).

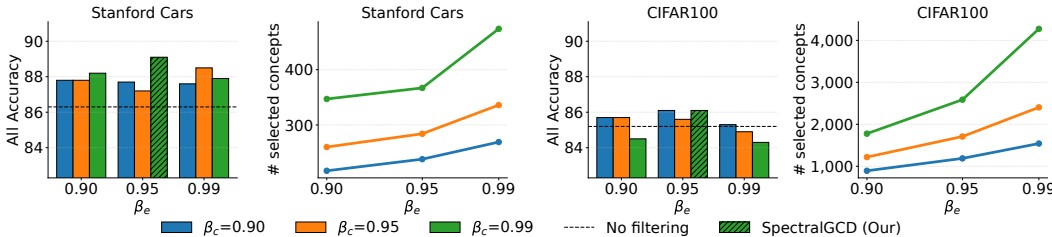

Figure 5: Ablation on thresholds $\beta_e$ and $\beta_c$. For each threshold we report *All* accuracy and the resulting number of selected concepts. The black dashed line denotes performance without Spectral Filtering; the hatched bar marks the chosen configuration used for our main results.

Only the unimodal SimGCD does not require a preparation phase. Our method achieves the highest accuracy while training faster than both GET and TextGCD, with training time comparable to the unimodal SimGCD. This highlights the efficiency of our approach, a desirable characteristic in realistic settings where discovery must be rerun as new unlabeled data arrives. Additional timing results on other datasets are reported in Appendix F.

## 5.3 ABLATIONS AND ANALYSES

**Forward/Reverse distillation ablations.** Table 2 reports the mean *Spearman correlation* between student and teacher cross-modal representations trained for different distillation variants (*FD+RD*: forward+reverse, *FD*: forward only, *RD*: reverse only, *No KD*: baseline). The Spearman correlation evaluates whether the student preserves the teacher's relative similarity structure, providing a rank-based perspective that is more robust than direct distance comparisons. Combining forward and reverse distillation yields the strongest alignment, reflected in higher correlations and *All* accuracy. All variants outperform the baseline, showing that KD effectively shapes the representation space, and improves downstream performance (see Appendix C).

Table 2: Spearman correlation and *All* accuracy between student and teacher representations on Stanford Cars varying distillation type.

| Distillation Loss | Spearman $\rho$ | All |
|---|---|---|
| FD + RD | $0.665 \pm 0.09$ | **89.1** |
| FD | $0.639 \pm 0.11$ | 86.0 |
| RD | $0.611 \pm 0.11$ | 87.5 |
| No KD | $0.487 \pm 0.15$ | 77.4 |

**Performance with Different Thresholding Values.** In Figure 5 we show the performance on Stanford Cars and CIFAR-100 when filtering the dictionary with different values of $\beta_e$ (threshold on eigenvalues) and $\beta_c$ (for the concept importance selection). Our default values are $\beta_e = 0.95$ and $\beta_c = 0.99$. We also consider no Spectral Filtering, using all concepts from the large dictionary. Spectral Filtering consistently improves results, with the effect being most pronounced on Stanford Cars. This highlights that carefully selecting concepts is especially beneficial for fine-grained datasets in which the number of selected concepts is of the same order of magnitude as the number of categories (200–450 concepts for 196 classes). On the coarse-grained CIFAR-100, the impact of filtering is more modest, with more concepts (1000–4000 concepts for 100 classes) being selected.

**Dictionary Choice Analysis.** Table 3 compares two concept dictionaries: the *Tags* used by TextGCD, a list of ~22K concepts arising from several benchmark datasets and used for reporting performance in Table 1, and the large-scale *OpenImages-v7* (Krasin et al., 2017) label set, which covers ~21K visual categories. We report TextGCD results using only these dictionaries, without the auxiliary LLM-generated *Attributes* dictionary, so that both methods operate under the same textual constraints.

These results show that *Tags* provide more relevant cues for category recognition on *New*, yielding consistently higher overall and novel-class accuracy, while OpenImages-v7 occasionally performs better on known classes. Our method improves over TextGCD in both settings, demonstrating more robustness to dictionary choice and consistently improving performance (see Appendix D for additional results and Table 13 for additional comparative results using the WordNet dictionary).

Table 3: Comparison of different dictionaries.

| Method | Dictionary | Stanford Cars | | | CIFAR100 | | |
|---|---|---|---|---|---|---|---|
| | | All | Old | New | All | Old | New |
| TextGCD* | OpenImagesV7 | 78.1 | 83.3 | 75.6 | 82.6 | 84.6 | 78.7 |
| TextGCD* | Tags | 86.2 | 91.2 | 83.8 | 84.3 | 84.8 | 83.3 |
| Ours | OpenImagesV7 | 85.8 | **93.8** | 82.0 | 84.9 | **87.6** | 79.4 |
| Ours | Tags | **89.1** | 92.6 | **87.4** | **86.1** | 87.2 | **83.9** |

**Choice of Teacher.** In Table 4 we show the effects of using different teacher models for both Spectral Filtering and distillation: the ViT-B/16 trained by OpenAI, the ViT-H/14 trained on LAION-2B (our and TextGCD's teacher for results in Table 1), and the ViT-H/14-QuickGELU pretrained on DFN-

Table 4: Comparison of different teachers.

| Teacher Model | CUB | | | CIFAR100 | | |
|---|---|---|---|---|---|---|
| | All | Old | New | All | Old | New |
| ViT-B/16 | 72.7 | 74.2 | 71.9 | 78.5 | 82.5 | 70.4 |
| H/14 (LAION-2B) | 79.2 | 80.4 | 78.5 | 86.1 | 87.2 | 83.9 |
| H/14-QuickGelu (DFN-5B) | **80.6** | **81.8** | **80.0** | **88.8** | **90.9** | **84.6** |

5B dataset (Fang et al., 2024), which has comparable number of parameters but bigger pretraining dataset. Through the use of Spectral Filtering and forward-reverse distillation losses, a stronger teacher can transfer higher-quality representations to the student, leading to higher performance (additional details are provided in Appendix E).

**Student Backbone Capacity.** To quantify the impact of the student architectures, in Table 5 we report results on CUB with the same student backbone as the teacher (i.e. ViT-H/14, LAION-2B). For a fair comparison, we report the original SimGCD and TextGCD results with the same architecture from TextGCD paper, marked by [†]. We reproduced methods marked by *. We observe that increasing student capacity leads to a small but consistent improvement in overall accuracy, indicating that our method can benefit from a stronger student backbone. The gains,

Table 5: Comparison of different students.

| Method (Backbone) | CUB | | |
|---|---|---|---|
| | All | Old | New |
| SimGCD (ViT-B/16)* | 71.1 | 75.6 | 68.8 |
| TextGCD (ViT-B/16)[†] | 76.6 | **80.6** | 74.7 |
| Ours (ViT-B/16) | **79.2** | 80.4 | **78.5** |
| SimGCD (ViT-H/14)[†] | 69.1 | 76.3 | 65.4 |
| TextGCD (ViT-H/14)[†] | 78.6 | 81.5 | 77.1 |
| Ours (ViT-H/14) | **80.0** | **82.9** | **78.6** |

however, are modest and the performance on New classes is essentially unchanged. Importantly, even when using the same high-capacity architecture as the teacher (ViT-H/14), our approach still surpasses SimGCD and TextGCD which also uses H/14 and achieves comparable performance to the teacher model with full knowledge of all class names. This suggests that the improvements are primarily driven by the contributions of SpectralGCD rather than by model scaling.

**Spectral Filtering on Different Splits** Table 6 reports results from additional experiments in which Spectral Filtering is computed only from Old samples (either all Old or only labeled Old). We note that performance remains close to the full version, indicating that SpectralGCD is robust to moderate distribution changes, since it works with a dictionary filtered using only partial knowledge of the full distribution of the dataset. Even so, if the distribution changes significantly and recomputing

Table 6: Spectral Filtering performance on CUB: Full dataset vs. Old (labeled and unlabeled) subsets.

| Spectral Filtering Data Split | CUB | | |
|---|---|---|---|
| | All | Old | New |
| Full Dataset | **79.2** | 80.4 | **78.5** |
| Only Old (Labeled + Unlabeled) | 77.8 | **80.8** | 76.4 |
| Only Old (Labeled) | 78.0 | 80.4 | 76.8 |

the full filtering step becomes the best alternative, the overall pipeline remains significantly cheaper than other multimodal GCD methods (see Figure 4 and Appendix F).

## 6 CONCLUSIONS

Generalized Category Discovery requires carefully balancing performance on *Old* labeled classes and generalization to *New* unlabeled classes, without sacrificing computational efficiency. We propose SpectralGCD, an efficient multimodal method that achieves state-of-the art performance across six benchmarks, improving results on both *Old* and *New* classes while significantly reducing computational overhead. By leveraging CLIP image-concept similarities as a unified cross-modal representation, SpectralGCD grounds parametric classifier learning in explicit semantics and reduces reliance on spurious visual correlations that cause overfitting on *Old* classes. Through Spectral Filtering, we retain the most relevant concepts from a large agnostic dictionary to build the cross-modal representation, while forward and reverse distillation further refine it during training.

**Limitations and Future Work.** SpectralGCD depends on the choice of teacher and dictionary (Table 3 and 4). It requires a teacher with sufficient domain knowledge and a dictionary with sufficient concept breadth for the downstream task. This limits robustness when domain coverage is incomplete. As future work, we aim to develop image-specific cross-modal representations to reduce reliance on the teacher and dictionary, further improving generalization.

## REPRODUCIBILITY STATEMENT

We have taken steps to ensure the reproducibility of our work. The full source code, along with scripts to reproduce all results in the paper, is publicly available. All experiments were performed on publicly available datasets, and details of model architectures, and main training hyper-parameters are given in the main paper with additional details included in the supplementary material (see Appendix A). To ensure the reproducibility of stochastic processes, such as weight initialization and dataset shuffling, we fix random seeds across all experiments reporting standard deviations in the appendix (see Table 7).

## ACKNOWLEDGMENTS

This work was partially supported by the AI4Debunk project (HORIZON-CL4-2023-HUMAN-01-CNECT grant n.101135757).

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

## APPENDIX A    IMPLEMENTATION DETAILS

We use the CLIP with image encoder ViT-B/16 trained by OpenAI (Radford et al., 2021), to ensure fairness with the rest of the multimodal methods in GCD. We train only the last transformer block of the visual encoder. The text encoder is left frozen and used only once to extract the student textual embeddings. The linear layer $W$ which maps the cross-modal representation has 768 as output dimension, to match the input size of both the classifier and MLP. The balance coefficient $\lambda$ is set to 0.35 as in Wen et al. (2023), while the thresholdings $\beta_e$ and $\beta_c$ are set to 0.95 and 0.99 respectively (see Figure 5). After Spectral Filtering, we append the *Old* class names to the filtered dictionary, as they are available under standard GCD protocol. We set the same CLIP's logit temperatures for both the teacher and the student ($\tau_t = \tau_s = 0.01$), while the $\tau$ temperature used for classification and contrastive objectives is fixed to 0.1, following previous works. We train the model for 200 epochs. We used two different learning rates with cosine annealing for the image encoder and the remaining learnable parameters. The visual backbone is fine-tuned with a learning rate of $5e - 3$, while for the classifier, MLP and linear layer learning we set it to 0.1. Training batch size is fixed at 128.

In the evaluation protocol, the clustering accuracy is computed by matching model predictions $\hat{y}_i$ to the ground-truth labels $y_i$ as:

$$\text{ACC} = \max_{p \in \mathcal{P}(\mathcal{Y}_u)} \frac{1}{|\mathcal{D}_u|} \sum_{i=1}^{|\mathcal{D}_u|} \mathbf{1}\{y_i = p(\hat{y}_i)\}, \tag{12}$$

where $\mathcal{D}_u$ is the unlabeled set and $\mathcal{P}(\mathcal{Y}_u)$ is the set of all label permutations for the unlabeled classes $\mathcal{Y}_u$. Accuracy is then reported for three subsets: **All**: all instances in $\mathcal{D}_u$; **Old**: instances in $\mathcal{D}_u$ whose labels belong to the labeled set $\mathcal{Y}_l$; and **New**: instances in $\mathcal{D}_u$ whose labels belong to the novel classes $\mathcal{Y}_u \setminus \mathcal{Y}_l$. All experiments are conducted on a single NVIDIA RTX 4090. Reported performance is averaged on three different seeds (see Table 7 for standard deviations). In Table 8 we include additional dataset details, regarding the samples count and splits used in our evaluation.

## APPENDIX B    BALANCE BETWEEN OLD AND NEW PERFORMANCE IN GCD

In this section, we discuss how unimodal approaches suffer from the *Old/New* tradeoff more severely than multimodal approaches. We then discuss how replacing image features with cross-modal features helps mitigating the gap between *Old* and *New* classes. Finally, we demonstrate that our cross-modal representation produces more compact clusters compared to relying only on visual features.

### B.1    TRADEOFFS IN UNIMODAL AND MULTIMODAL APPROACHES

Figure 6 compares the *relative accuracy* (i.e. the ratio between the *New* and *Old* accuracy) of different unimodal and multimodal methods across Stanford Cars (left) and ImageNet-100 (right). Unimodal methods suffer from a more pronounced *New/Old* tradeoff, with substantially lower relative accuracy. In contrast, multimodal methods, particularly TextGCD and SpectralGCD, achieve higher relative accuracy, narrowing this gap.

### B.2    CROSS-MODAL REPRESENTATION VERSUS IMAGE FEATURES

To better understand the contribution of our cross-modal representation, we compare our method trained using either the cross-modal representation or image features in input to a parametric classifier, denoted as *Cross-Modal* and *Image features* in Table 9 respectively. In the *Image Features*

Table 7: Mean and standard deviation of Spectral-GCD across the six evaluated benchmarks.

| Dataset | All | Old | New |
|---|---|---|---|
| CUB | $79.2 \pm 0.9$ | $80.4 \pm 0.4$ | $78.5 \pm 1.1$ |
| Stanford Cars | $89.1 \pm 0.9$ | $92.6 \pm 1.1$ | $87.4 \pm 1.6$ |
| FGVC-Aircraft | $63.0 \pm 1.6$ | $66.1 \pm 1.5$ | $61.4 \pm 2.2$ |
| CIFAR10 | $98.5 \pm 0.0$ | $96.7 \pm 0.1$ | $99.4 \pm 0.0$ |
| CIFAR100 | $86.1 \pm 0.8$ | $87.2 \pm 0.3$ | $83.9 \pm 2.3$ |
| ImageNet-100 | $93.4 \pm 0.7$ | $96.1 \pm 0.0$ | $92.0 \pm 1.1$ |

Table 8: Statistics of all the evaluated benchmark datasets.

| Dataset | Labelled | | Unlabelled | |
|---|---|---|---|---|
| | Images | Classes | Images | Classes |
| CUB | 1.5K | 100 | 4.5K | 200 |
| Stanford Cars | 2.0K | 98 | 6.1K | 196 |
| FGVC-Aircraft | 1.7K | 50 | 5.0K | 100 |
| CIFAR10 | 12.5K | 5 | 37.5K | 10 |
| CIFAR100 | 20.0K | 80 | 30.0K | 100 |
| ImageNet-100 | 31.9K | 50 | 95.3K | 100 |

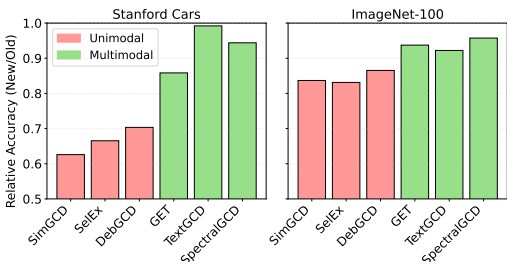 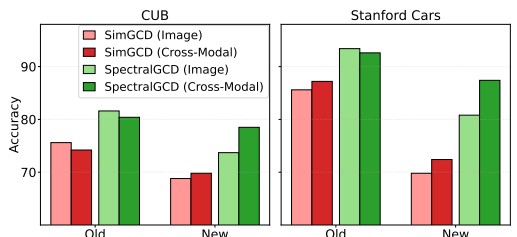

Figure 6: Relative accuracy (*New/Old*) categories for Stanford Cars and ImageNet-100. Multimodal methods consistently reduce the *New/Old* gap compared to unimodal baselines.

Figure 7: Performance comparison between SimGCD trained with CLIP backbone and SpectralGCD when training the classifier on either image features (Image) or cross-modal representations (Cross-Modal).

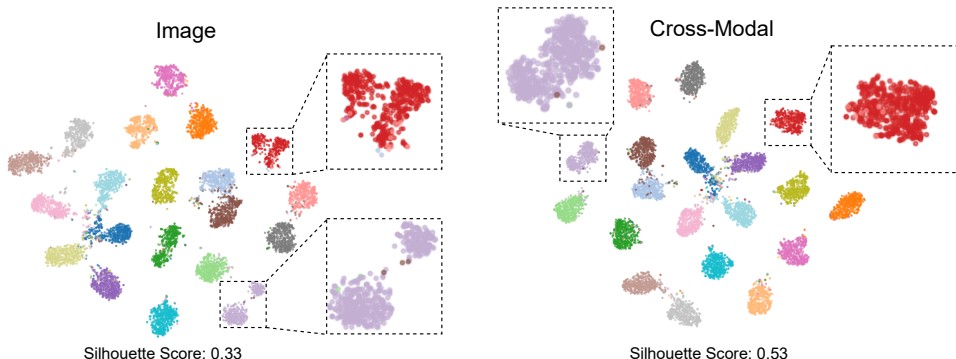

Figure 8: t-SNE comparison across all the 20 CIFAR100 *New* classes, between SpectralGCD image features extracted when training the classifier only on image features (**left**), and cross-modal features when training SpectralGCD classifier on cross-modal features, following the proposed approach (**right**). Silhouette scores show the efficacy of our cross-modal based approach.

variant, the full SpectralGCD pipeline is retained, including spectral filtering and forward/reverse distillation, and only the classifier input is swapped to image features. The ablation results on CUB, Stanford Cars, and CIFAR100 show that training a classifier using cross-modal representations consistently achieves the best overall performance, offering a stronger balance by generalizing well to *New* categories while maintaining competitive accuracy on *Old* ones. In contrast, the image-only approach tend to overfit to *Old* categories. This trend aligns with the observations depicted in Figures 3 and 7, which shows that a similar pattern is observed when training SimGCD with image or cross-modal features. For the SimGCD (Cross-Modal) variant, the dictionary is filtered using the student itself, as SimGCD does not receive teacher distillation that would otherwise motivate teacher-based filtering.

To further illustrate the differences between image-only features and cross-modal representations, in Figure 8 we present a t-SNE comparison of the learned embeddings across all the 20 CIFAR100 *New* classes available in the evaluation set. The visualization contrasts SpectralGCD image features when the classifier is trained only on visual inputs (left), with cross-modal features obtained when the SpectralGCD classifier is trained under our proposed cross-modal supervision (right).

The cross-modal representation produces more compact clusters compared to the image-only features, reflecting the stronger discriminative power of the cross-modal trained model to learn a more structured embedding space. To quantitatively validate this observation, we report silhouette scores, which measure intra-class compactness and inter-class separability. Consistently, the cross-modal representations achieve higher silhouette scores, confirming their stronger discriminative power compared to image-only features.

Table 9: Ablation comparing cross-modal representation to image representation parametric classifier inputs. Cross-modal representation achieve better generalization to *New* categories, while retaining good performance on *Old*.

| Configuration | CUB | | | Stanford Cars | | | CIFAR100 | | |
|---|---|---|---|---|---|---|---|---|---|
| | All | Old | New | All | Old | New | All | Old | New |
| Cross-modal | **79.2** | 80.4 | **78.5** | **89.1** | 92.6 | **87.4** | **86.1** | 87.2 | **83.9** |
| Image Features | 76.3 | **81.6** | 73.7 | 84.9 | **93.4** | 80.8 | 85.0 | **88.7** | 77.8 |

Table 10: Ablation on knowledge distillation strategies. We compare the usage of only forward distillation (FD), only reverse (RD), their combination (FD+RD), and no distillation (No KD). Employing both forward and reverse distillation yields the best performance, or performance comparable to the best, across the datasets we evaluated.

| Distillation | CUB | | | Stanford Cars | | | CIFAR100 | | | Aircraft | | |
|---|---|---|---|---|---|---|---|---|---|---|---|---|
| | All | Old | New | All | Old | New | All | Old | New | All | Old | New |
| FD+RD | 79.2 | 80.4 | **78.5** | **89.1** | 92.6 | **87.4** | **86.1** | 87.2 | **83.9** | **63.0** | **66.1** | **61.4** |
| RD | **79.3** | **81.0** | 78.4 | 87.5 | **93.3** | 84.6 | 85.5 | 87.4 | 81.7 | 61.4 | 63.8 | 60.2 |
| FD | 77.0 | 79.3 | 75.9 | 86.0 | 92.3 | 82.9 | 84.5 | 86.7 | 80.1 | 59.7 | 60.1 | 59.5 |
| No KD | 71.6 | 75.7 | 69.5 | 77.4 | 86.2 | 73.1 | 80.0 | 83.8 | 72.5 | 52.0 | 56.9 | 49.6 |

## APPENDIX C  EFFECTS OF DIFFERENT DISTILLATIONS

We analyze the impact of forward distillation (FD), reverse distillation (RD), and their combination on refining the student's cross-modal representation (Table 10). RD proves especially effective at retaining previously acquired knowledge, yielding the highest accuracy on *Old* classes. In contrast, FD better facilitates adaptation to *New* categories. Combining FD and RD strikes the most balanced trade-off, achieving strong generalization to *New* categories while maintaining competitive results on *Old* ones,leading to the overall best performance across datasets. We also evaluate training without any distillation, i.e., without leveraging the teacher's guidance. This leads to a marked drop in performance, confirming the importance of distillation for enriching the target model.

We also note that in some cases (like for CUB), using only the reverse term actually helps in achieving better performance compared to using both directions. This is because while forward distillation forces the student to match high probability concepts of the teacher, the reverse actually does the opposite: it tells where not to focus and reduce the probability of the highly unlikely concepts.

### C.1  STUDENT-TEACHER ALIGNMENT

In Table 11 we report the Spearman correlation and the mean P-value on CUB, Stanford Cars, and CIFAR100. We use the Spearman correlation to assess the alignment between teacher and student cross-modal representations. Unlike raw distance measures, it evaluates whether the student preserves the relative ordering of similarities produced by the teacher, a rank-based perspective that is more robust and better reflects the semantic structure of the representation space.

We compute the Spearman correlation for models trained under different distillation schemes: using both forward and reverse (FD+RD), using only forward distillation (FD), only reverse distillation (RD) and employing not distillation at all (No KD). Using both forward and reverse distillation consistently yields the highest Spearman correlation between student and teacher features, indicating stronger student–teacher alignment. Importantly, this stronger alignment directly translates into better performance (as shown in Table 10), confirming that maintaining a high alignment with the teacher is beneficial for effective knowledge transfer.

## APPENDIX D  ANALYSIS OF DICTIONARY CHOICE

Table 12 compares two dictionary sources: *Tags*, curated from benchmark datasets and used in the main results, and the large-scale *OpenImages-v7* (Krasin et al., 2017) which spans thousands of cat-

Table 11: Spearman correlation between student and teacher cross-modal representations for varying distillation strategiy on CUB, Stanford Cars, and CIFAR100. We compare the usage of only forward distillation (FD), only reverse (RD), their combination (FD+RD), and no distillation (No KD). We report the mean $\rho$ and the $p$-value.

| Distillation Loss | CUB | | Stanford Cars | | CIFAR100 | |
|---|---|---|---|---|---|---|
| | Spearman $\rho$ | $p$-value | Spearman $\rho$ | $p$-value | Spearman $\rho$ | $p$-value |
| FD + RD | $0.7113 \pm 0.09$ | $7.0\mathrm{e}^{-10}$ | $0.6654 \pm 0.09$ | $1.1\mathrm{e}^{-6}$ | $0.5809 \pm 0.07$ | $2.2\mathrm{e}^{-28}$ |
| FD | $0.6819 \pm 0.09$ | $2.4\mathrm{e}^{-5}$ | $0.6391 \pm 0.11$ | $1.3\mathrm{e}^{-5}$ | $0.5648 \pm 0.07$ | $2.9\mathrm{e}^{-25}$ |
| RD | $0.6762 \pm 0.09$ | $1.6\mathrm{e}^{-8}$ | $0.6111 \pm 0.11$ | $4.2\mathrm{e}^{-5}$ | $0.5542 \pm 0.08$ | $1.1\mathrm{e}^{-12}$ |
| No KD | $0.5314 \pm 0.12$ | $3.7\mathrm{e}^{-3}$ | $0.4873 \pm 0.15$ | $8.9\mathrm{e}^{-3}$ | $0.3040 \pm 0.15$ | $1.6\mathrm{e}^{-2}$ |

Table 12: Performance comparison using different dictionaries (*OpenImagesV7*, *Tags*) for both our method and TextGCD. Results on fine-grained benchmarks show that the Tags dictionary benefits both methods, underscoring the importance of dictionary selection for generalization.

| Method | Dictionary | CUB | | | Stanford Cars | | | CIFAR100 | | |
|---|---|---|---|---|---|---|---|---|---|---|
| | | All | Old | New | All | Old | New | All | Old | New |
| TextGCD | OpenImagesV7 | 64.2 | 64.2 | 64.2 | 78.1 | 83.3 | 75.6 | 82.6 | 84.6 | 78.7 |
| TextGCD | Tags | 73.8 | 75.9 | 72.7 | 86.2 | 91.2 | 83.8 | 84.3 | 84.8 | 83.3 |
| Ours | OpenImagesV7 | 77.3 | **81.1** | 75.5 | 85.8 | **93.8** | 82.0 | 84.9 | **87.6** | 79.4 |
| Ours | Tags | **79.2** | 80.4 | **78.5** | **89.1** | 92.6 | **87.4** | **86.1** | 87.2 | **83.9** |

egories. Both contain a similar number of entries. Here, we report TextGCD results using only these dictionaries, excluding the auxiliary LLM-generated *Attributes* dictionary, to ensure that both methods operate under the same textual constraints. Results on CUB are consistent with those observed on the other datasets: *Tags* provide more relevant cues for novel category recognition, yielding higher overall and novel-class accuracy, whereas OpenImages-v7 occasionally performs better on known classes. Overall, our method demonstrates greater robustness to dictionary variations.

To further evaluate the robustness of our method even when using non vision-centric dictionaries, in Table 13 we report results from additional experiments for SpectralGCD and TextGCD using WordNet, a general linguistic ontology that includes many concepts without visual grounding. Specifically, we extracted the first lemma of each WordNet synset to create a dictionary covering all lexicalized concepts present in WordNet. We further compare TextGCD and SpectralGCD with GET, which however does not require any additional dictionary to work.

SpectralGCD remains robust even with a generic dictionary that is not vision-centric, outperforming TextGCD under the same conditions and still outperforming GET. This shows that our method can extract useful semantic structure even when the dictionary is not curated from visual benchmark concepts. As expected, Tags (a domain-aligned dictionary of visual concepts) yields the best performance. These results show that SpectralGCD is not limited to vision-centric dictionaries and can operate effectively with generic concept sets such as WordNet.

We further examine scalability. While the method involves computing an $M \times M$ covariance matrix, in practice this operation is performed only once per dictionary and can be efficiently managed using low-rank approximations. To assess scalability beyond the 20K-concept dictionaries used in our main experiments, we conducted additional tests with a substantially larger dictionary of roughly 60K concepts (based on WordNet, as shown in Table 13). For this setting, we used a low-rank approximation (i.e., computing only the top-$K$ eigenvectors needed by our method rather than the full spectrum). This reduces both compute time and memory. The memory requirement in particular goes from $O(M^2)$ to approximately $O(MK)$, making spectral filtering feasible even at this scale on a single GPU (an NVIDIA RTX 4090 in our case). Thus, low-rank approximations keep the method scalable, and we verify feasibility up to 60K concepts.

## APPENDIX E    IMPACT OF TEACHER SELECTION

In Table 14, we report the impact of using different teachers for both Spectral Filtering and distillation. We consider three candidates: the ViT-B/16 released by OpenAI (also used as our stu-

Table 13: Performance comparison between using the Tags dictionary versus using WordNet dictionary. We report results for both SpectralGCD and TextGCD.

| Method | Dictionary | CUB | | | Stanford Cars | | |
|---|---|---|---|---|---|---|---|
| | | All | Old | New | All | Old | New |
| GET | N/A | 77.0 | 78.1 | 76.4 | 78.5 | 86.8 | 74.5 |
| TextGCD | WordNet | 69.8 | 76.2 | 66.6 | 63.6 | 84.7 | 53.4 |
| TextGCD | Tags | 69.9 | 74.9 | 67.3 | 86.2 | 91.2 | 83.8 |
| Ours | WordNet | 77.1 | 79.9 | 75.7 | 83.0 | **93.5** | 78.0 |
| Ours | Tags | **79.2** | **80.4** | **78.5** | **89.1** | 92.6 | **87.4** |

Table 14: Comparison of teacher variants (ViT-B/16 architecture released by OpenAI, ViT-H/14 (LAION-2B) and ViT-H/14-QuickGelu (DFN-5B)). We evaluate the influence of different teacher models when applied both for Spectral Filtering and during the distillation of cross-modal representations to the student.

| Teacher Model | CUB | | | Stanford Cars | | | CIFAR100 | | |
|---|---|---|---|---|---|---|---|---|---|
| | All | Old | New | All | Old | New | All | Old | New |
| ViT-B/16 | 72.7 | 74.2 | 71.9 | 75.1 | 84.7 | 70.4 | 78.5 | 82.5 | 70.4 |
| H/14 (LAION-2B) | 79.2 | 80.4 | 78.5 | 89.1 | 92.6 | **87.4** | 86.1 | 87.2 | 83.9 |
| H/14-QuickGELU (DFN-5B) | **80.6** | **81.8** | **80.0** | **89.5** | **94.4** | 87.2 | **88.8** | **90.9** | **84.6** |

dent), the ViT-H/14 trained on LAION-2B (the main teacher in our reported results), and the ViT-H/14-QuickGELU[2] trained on the DFN-5B dataset (Fang et al., 2024), where Data Filtering Networks were employed to curate 5B image–text pairs from a pool of 43B. Notably, the two ViT-H/14 teachers share the same backbone size but differ in their pretraining data.

Results in Table 14 demonstrate that stronger teachers consistently provide richer semantic cross-modal signals. These signals not only enhance the quality of the distilled student representations but also improve the reliability and effectiveness of Spectral Filtering. This reinforces our main findings (Table 4): leveraging teachers with higher capacity and broader pretraining yields more discriminative and better-aligned representations. Such representations, in turn, facilitate a more faithful transfer of semantic knowledge across modalities and ultimately translate into superior Spectral-GCD performance.

## APPENDIX F    TIMING ANALYSIS

In Figure 9, we report total training times across four datasets, including the preparation phase time for multimodal methods. All methods are trained with the same fixed number of epochs, as in their original implementations and following standard GCD protocol. The trends observed on CUB in Figure 4 are consistent for Stanford Cars, CIFAR100, and ImageNet-100: our method trains substantially faster than the multimodal competitors GET and TextGCD, while remaining comparable to the unimodal baseline SimGCD, which does not require a preparation phase as multimodal methods. This efficiency arises from training only the student's visual encoder, while freezing the text encoder. The latter is used once at the beginning of training to extract dictionary text embeddings, after which no further forward passes are required. The efficiency particularly beneficial in realistic scenarios where category discovery must be performed repeatedly as new data streams in. This makes the method especially well-suited for dynamic environments with incremental category growth, where fast adaptation is essential.

## APPENDIX G    FINE-TUNING THE TEXT ENCODER

In Table 15 we show the effects of fine-tuning solely the visual encoder versus fine-tuning both visual and text encoders. In SpectralGCD, we intentionally keep the text encoder frozen because the text embeddings define a stable semantic basis for measuring image–concept similarities. Fine-tuning both encoders causes them to drift together, breaking this anchor. While OLD categories see

---

[2]https://huggingface.co/apple/DFN5B-CLIP-ViT-H-14

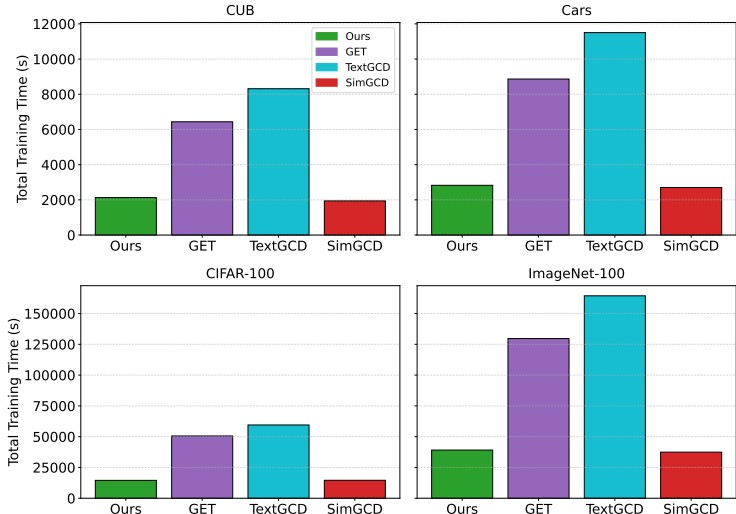

Figure 9: Total training time on CUB, Stanford Cars, CIFAR100, and ImageNet-100, including the preparation phase for multimodal methods (Ours, GET, and TextGCD). SimGCD is the only method that do not require a preparation phase.

Table 15: Proposed SpectralGCD Vision only fine-tuning vs Vision + Text fine tuning.

|             | **CUB** | | | **Stanford Cars** | | |
|-------------|------|------|------|------|------|------|
| **Method**  | All  | Old  | New  | All  | Old  | New  |
| Vision only | **79.2** | 80.4 | **78.5** | **89.1** | **92.6** | **87.4** |
| Vision + Text | 77.6 | **81.0** | 75.9 | 87.7 | 91.3 | 86.0 |

marginal gains on CUB, this comes at the expense of a degradation in discovering NEW categories. This behavior aligns with recent findings on vision-language model adaptation (Shu et al., 2023; Gao et al., 2024), which demonstrate that keeping the text encoder frozen preserves the pretrained semantic structure essential for generalization. In our setting, the frozen text space provides the semantic foundation for spectral filtering, enabling discovery of novel categories through their relationship to known concepts. Fine-tuning causes the text encoder to overfit to labeled OLD categories, losing the general semantic structure needed in GCD.

## APPENDIX H  ABLATION ON DIFFERENT OLD/NEW IMBALANCE.

To assess sensitivity to class imbalance in the unlabeled set, Table 16 show additional experiments on CUB by varying the proportion of New classes between 20% and 80%. With 20% New, SpectralGCD maintains strong performance and continues to outperform unimodal and multimodal approaches. The 80% New setting is much more challenging. Still SpectralGCD improves

Table 16: Sensitivity to Old/New imbalance on the CUB dataset.

|         | **CUB (20% New Classes)** | | | **CUB (80% New Classes)** | | |
|---------|------|------|------|------|------|------|
| **Method** | All | Old | New | All | Old | New |
| SimGCD  | 77.4 | 81.1 | 69.8 | 54.5 | 41.1 | 56.2 |
| GET     | 80.7 | 82.8 | **76.5** | 61.3 | 49.3 | 62.8 |
| TextGCD | 78.0 | 79.0 | 75.9 | **71.6** | **65.5** | **72.4** |
| Ours    | **82.5** | **85.8** | 75.8 | 68.1 | 59.7 | 69.1 |

upon SimGCD and GET by a large margin. TextGCD performs best in this imbalanced scenario since it directly leverages the Teacher's text assignments, giving it a strong initialization when the vast majority of unlabeled samples come from unseen categories. This advantage disappears once the distribution becomes less extreme, as represented by the performance on the 20% New settings and the standard performance in the 50/50 split from Table 1.

Table 17: Top-ranked discriminative concepts and common background concepts (with relative ranks) obtained using Spectral Filtering on the CUB dataset with a WordNet dictionary.

| Top Discriminative Concepts | | Common Background Concepts | |
|---|---|---|---|
| Rank | Concept | Rank | Concept |
| 1 | Tern | 354 | Beak |
| 2 | Warbler | 9,950 | Wing |
| 3 | Woodpecker | 9,700 | Branch |
| 4 | Vesper Sparrow | 21,623 | Leg |
| 5 | Indigo Bunting | 23,420 | Grass |
| 6 | Vermilion Flycatcher | 25,964 | Tree |
| 7 | Red-headed Woodpecker | 29,177 | Bush |
| 8 | Rusty Blackbird | 36,945 | Leaf |
| 9 | Evening Grosbeak | 43,612 | Sky |
| 10 | Whip-poor-will | 44,501 | Cloud |

Table 18: Comparison of linear probing performance (Left) and image-to-image retrieval (Right) between Image Features and our Cross-modal representations on CUB and Stanford Cars datasets.

| | Accuracy (%) | | | | CUB | | Cars | |
|---|---|---|---|---|---|---|---|---|
| Feature Type | CUB | Cars | Feature Type | | mAP | R@1 | mAP | R@1 |
| Image Features | **88.5** | **95.1** | Image Features | | 56.3 | 80.4 | 61.8 | 88.9 |
| Cross-modal representations | 87.6 | 94.9 | Cross-modal representations | | **62.1** | **81.7** | **75.4** | **91.4** |

## APPENDIX I   DISCRIMINATIVENESS OF THE CROSS-MODAL REPRESENTATION

To qualitatively show the effectiveness spectral filtering, in Table 17 we report the top-10 selected concepts on the CUB dataset (a dataset containing 200 fine-grained bird species) obtained when filtering the WordNet dictionary ( 60K concepts) using the concept importance score defined in Eq. (8).

Finally, to provide more articulated, in Table 18 we show empirical evidence of the discriminative power of the selected concepts with two additional experiments, comparing the image features and our filtered cross-modal representations computed using the frozen Teacher ViT-H/14 on CUB and Stanford Cars. In the first experiment ( Table 18 (Left)), we follow the standard linear probing evaluation protocol. We keep the encoder frozen and train a linear classifier on the full training split, supposing access to all labels. In the second experiment ( Table 18 (Right)), we run a standard image-to-image retrieval evaluation considering images of the same class as positive examples.

Results show that the cross-modal representation (even in a smaller representation space of 263 concepts for CUB and 377 for cars) are as linearly separable as the original image features (of embedding dimension 1024), with an accuracy of 87.6% compared to 88.5%. More interestingly, our filtered cross-modal features are better calibrated for the harder, unsupervised task of image-to-image retrieval, improving mAP by nearly 6% on CUB and 14% on Cars over the image features. This showcases the efficacy of our approach in selecting task-relevant concepts.

