# OpenReview forum: "SpectralGCD: Spectral Concept Selection and Cross-modal Representation Learning for Generalized Category Discovery"
_ICLR.cc/2026/Conference — ICLR 2026 Poster_

### Official Review · Reviewer_hK7S · 2025-10-28

**Soundness:** 3
**Presentation:** 3
**Contribution:** 3
**Rating:** 4
**Confidence:** 5

**Summary:**

The paper proposes SpectralGCD, a multimodal approach for Generalized Category Discovery (GCD) that leverages CLIP’s cross-modal image–concept similarities as a unified representation. Instead of treating visual and textual modalities independently, SpectralGCD represents each image as a mixture over a large task-agnostic concept dictionary, which is then filtered via a novel Spectral Filtering mechanism based on eigendecomposition of a cross-modal covariance matrix derived from a frozen teacher model. The method further employs forward and reverse knowledge distillation to preserve semantic fidelity during student training. Evaluated across six benchmarks, SpectralGCD achieves state-of-the-art or competitive performance with significantly lower computational cost than existing multimodal GCD methods, while maintaining efficiency comparable to unimodal baselines.

**Strengths:**

(1) The core idea of using CLIP’s cross-modal similarities as a sufficient representation for GCD is both conceptually elegant and practically effective, grounding classification in explicit semantics and reducing overfitting to spurious visual cues.
(2) Spectral Filtering provides an automated, unsupervised way to prune irrelevant concepts from a large dictionary without relying on LLM-generated descriptions or manual curation, improving both representation quality and computational efficiency.
(3) The combination of forward and reverse knowledge distillation ensures strong alignment between student and teacher cross-modal representations, which is empirically validated through Spearman correlation and ablation studies.
(4) The method achieves state-of-the-art results across diverse benchmarks (fine- and coarse-grained) while being significantly faster to train than other multimodal approaches like GET and TextGCD, making it suitable for real-world deployment scenarios requiring repeated discovery.

**Weaknesses:**

(1) It is hard to figure out the novelty as there is many works that constructs hierarchal fine-grained knowledge when performing tasks. Also, the paper assumes access to a “Tags” dictionary derived from benchmark datasets, but it is unclear how generalizable this dictionary is to truly out-of-domain tasks (e.g., medical or satellite imagery). While OpenImages-v7 is tested, both dictionaries are still vision-centric and curated from existing classification datasets. Could the authors clarify whether SpectralGCD would still perform well with a generic, non-vision-specific concept set (e.g., WordNet or Wikipedia titles), and what minimum coverage or semantic alignment is required between the dictionary and the target domain?
(2) Spectral Filtering relies on computing the full cross-modal covariance matrix G ∈ ℝ^{M×M}, where M is the dictionary size (~20K). For very large dictionaries (e.g., 100K+ concepts), this becomes memory-prohibitive (O(M²) storage). The paper mentions efficiency but does not discuss scalability limits of Spectral Filtering. Did the authors explore approximations (e.g., randomized SVD, Nyström) for larger M, and what is the practical upper bound on dictionary size given current GPU memory constraints?
(3) The distillation loss uses softmax-normalized similarities σ(zˆi) and σ(zˆi∗), but CLIP’s original logit scaling already includes a temperature τ. The paper sets τ = 0.01 for both teacher and student (Appendix A), yet the distillation loss applies another softmax. This may over-smooth or distort the relative concept rankings. Could the authors justify this design choice and provide ablation results comparing raw cosine similarities vs. softmax-normalized logits in the distillation objective?
(4) The student only fine-tunes the last transformer block of ViT-B/16, while the teacher is ViT-H/14. This architectural mismatch raises questions about the fairness of distillation: the student has far fewer parameters and less capacity. Would the performance gap between SpectralGCD and TextGCD shrink if both used the same backbone size? Also, why not use a ViT-B/16 teacher for a more direct comparison of the representation learning strategy alone?
(5) The evaluation protocol follows standard GCD practice, but all benchmarks assume that the unlabeled set contains a known split of Old and New classes (e.g., 50/50 in CIFAR100). How sensitive is SpectralGCD to imbalanced Old/New ratios in the unlabeled data? For instance, if New classes dominate (>80%), does the method still avoid collapsing New clusters into Old prototypes, and how does entropy regularization interact with such shifts?
(6) The paper claims that cross-modal representations reduce overfitting to Old classes, but Figure 3 and Table 7 show that on Stanford Cars, the “Image Features” variant actually achieves higher Old accuracy (93.4 vs. 92.6) than the cross-modal version. This contradicts the stated benefit. Could the authors explain this anomaly and clarify under what conditions cross-modal representations might sacrifice Old performance for New gains—or vice versa?
(7) The preparation phase for SpectralGCD (194s on CUB) includes precomputing teacher representations and performing Spectral Filtering. However, if new unlabeled data arrive incrementally (as mentioned in the introduction), does the entire filtering step need to be recomputed? If so, this could undermine the claimed efficiency in dynamic settings. Please clarify whether the filtered dictionary Cˆ is fixed after initial filtering or must be updated with new data.
(8) The reverse distillation term L_rd = −σ(zˆi) log σ(zˆi∗) penalizes the student for assigning high probability to concepts the teacher deems unlikely. However, if the teacher itself is biased or misaligned with the true semantics of a novel class (e.g., mislabeling a “sparrow” as “eagle”), wouldn’t reverse distillation reinforce this error? How robust is the method to teacher mistakes, especially on fine-grained novel categories where even strong CLIP models struggle?

**Questions:**

Please see Weakness.

---

> ### Author Response · Authors · 2025-11-22
> **Responses to Reviewer hK7S (1/3)**
>
> We thank the reviewer for their positive feedback and constructive criticism of our work. We appreciate their recognition of the elegance and effectiveness of leveraging CLIP’s cross-modal similarities for GCD, of our automated approach of Spectral Filtering to enhance representation quality and efficiency, and of the state-of-the-art results achieved by SpectralGCD across a diverse set of fine- and coarse-grained benchmarks. Below we respond to specific points raised in the initial review.
>
> ## On the task-agnosticity of our concept dictionary (W1)
>
> To directly address this concern, we conducted additional experiments using WordNet -- a broad, generic, non-vision-specific lexical ontology. In particular, we extracted the first lemma of each synset, creating a dictionary covering all lexicalized concepts present in WordNet. We repeated the experiments also for TextGCD using these dictionaries:
>
> |Method (Dictionary)|CUB ALL|CUB OLD|CUB NEW|Cars ALL|Cars OLD|Cars NEW|
> |-|-|-|-|-|-|-|
> |GET (N/A)|77.0|78.1|76.4|78.5|86.8|74.5|
> |TextGCD (WordNet)|69.8|76.2|66.6|63.6|84.7|53.4|
> |TextGCD (Tags)|69.9|74.9|67.3|86.2|91.2|83.8|
> |**Ours (WordNet)**|77.1|79.9|75.7|83.0|**93.5**|78.0|
> |**Ours (Tags)**|**79.2**|**80.4**|**78.5**|**89.1**|92.6|**87.4**|
> | |
>
> SpectralGCD remains robust even with a generic dictionary that is not vision-centric, outperforming TextGCD under the same conditions and still outperforming GET. This shows that our method can extract useful semantic structure even when the dictionary is not curated from visual benchmark concepts. As expected, Tags (domain-aligned) yields the best performance. These results show that SpectralGCD is not limited to vision-centric dictionaries and can operate effectively with generic concept sets such as WordNet or potentially other large knowledge sources. We added this experiment in **Table 13, Appendix D**.
>
> ## On the scalability of Spectral Concept Filtering (W2)
>
> We appreciate the reviewer's question regarding the scalability of Spectral Filtering. While the method involves an $M\times M$ covariance matrix, in practice this computation is required only once per dictionary, and can be efficiently handled using low-rank approximations. To evaluate scalability beyond the 20K-concept dictionaries used in our experiments, we conducted additional tests using a substantially larger dictionary (~60K concepts, using WordNet as shown above). For this setting, we used a low-rank approximation (i.e., computing only the top-$K$ eigenvectors needed by our method rather than the full spectrum). This reduces both compute time and memory. The memory requirement in particular goes from $O(M^2)$ to approximately $O(MK)$, making spectral filtering feasible even at this scale on a single GPU (e.g., RTX 4090).
>
> Moreover, since spectral filtering is an offline preprocessing step and need not run on the GPU, larger dictionaries can also be processed on CPU-only machines using the same low-rank approximation, at the cost of longer runtime that needs to be computed only one time, without affecting the training pipeline.
>
> Thus, while storing the full $M\times M$ matrix becomes impractical at very large M, the method scales well in practice when using standard low-rank approximations, and our experiments confirm feasibility up to ~60K concepts without modification to the learning pipeline.
>
> We added this discussion to **Appendix D, lines 1011-1020**.
>
> ## On the distillation loss and over-smoothing (W3)
>
> Although CLIP applies a temperature $\tau$ when computing the softmax over its cosine similarity scores before computing the contrastive loss during pretraining, but these values output by CLIP *not* probabilities but rather unbounded similarity scores. Since our distillation objective is formulated as a KL divergence, we require a distribution over concepts, which naturally involves applying a softmax to both teacher and student cosine similarity scores during distillation. The softmax is thus applied only once.
>
> To address the reviewer's concern about potential smoothing or distortion, we conducted the requested ablation by replacing our KL term with a distillation loss directly on the raw cosine similarities (L2):
>
> |Method|CUB ALL|CUB OLD|CUB NEW|
> |-|-|-|-|
> |**Ours** (KL)|**79.2**|**80.4**|**78.5**|
> |Raw L2 (no softmax)|70.8|74.9|68.8|
> | |
>
> Using the raw logits leads to a substantial drop in performance, particularly on NEW classes (−9.7%). This confirms that softmax normalization does not harm the ranking structure; rather, it provides a more stable and informative signal for distribution matching.

---

> ### Author Response · Authors · 2025-11-22
> **Responses to Reviewer hK7S (2/3)**
>
> ## On architectural mismatch and distillation fairness (W4)
>
> We agree that controlling for model capacity is important in evaluating the contribution of the proposed distillation mechanism. Note that for all experiments in the paper *the teacher and student networks used for TextGCD and SpectralGCD are identical* (ViT-B16 and ViT-H14, respectively). First, regarding fairness of the teacher–student comparison: our paper already includes a teacher capacity ablation (Tables 4 and 14) in which we evaluate teachers of different sizes (ViT-B/16, ViT-H/14 LAION-2B, and ViT-H/14 DFN-5B). To directly address the reviewer's concern, we additionally trained a high-capacity student matching the teacher's architecture (ViT-H/14):
>
> |Method (Backbone)|CUB ALL|CUB OLD|CUB NEW|
> |-|-|-|-|
> |TextGCD (ViT-B/16)|76.6|80.6|74.7|
> |**Ours** (ViT-B/16)|79.2|80.4|78.5|
> |SimGCD (ViT-H/14)*|69.1|76.3|65.4|
> |TextGCD (ViT-H/14)*|78.6|81.5|77.1|
> |**Ours** (ViT-H/14)|**80.0**|**82.9**|**78.6**|
> | |
>
> Even with an H/14 backbone, our method still clearly outperforms SimGCD-H/14 and TextGCD-H/14, which demonstrates that the gains do not come merely from using a smaller or larger student, but from the proposed strategy. We added these new results in **Table 5** in the revised main paper.
>
> ## On imbalance between Old and New classes (W5)
>
> To assess sensitivity to class imbalance in the unlabeled set, we conducted additional experiments on CUB by varying the proportion of New classes between 20% and 80%, following the reviewer's suggestion:
>
> ### Sensitivity to Old/New Imbalance on CUB (20% New Classes)
>
> |Method|CUB ALL|CUB OLD|CUB NEW|
> |-|-|-|-|
> |SimGCD|77.4|81.1|69.8|
> |GET|80.7|82.8|**76.5**|
> |TextGCD|78.0|79.0|75.9|
> |**Ours**|**82.5**|**85.8**|75.8|
> | |
>
> ### Sensitivity to Old/New Imbalance on CUB (80% New Classes)
>
> |Method|CUB ALL|CUB OLD|CUB NEW|
> |-|-|-|-|
> |SimGCD|54.5|41.1|56.2|
> |GET|61.3|49.3|62.8|
> |TextGCD|**71.6**|**65.5**|**72.4**|
> |**Ours**|68.1|59.7|69.1|
> | |
>
> With 20% New, SpectralGCD maintains strong performance and continues to outperform unimodal and multimodal approaches. The 80% New setting is much more challenging. Still SpectralGCD improves upon SimGCD and GET by a large margin. TextGCD performs best in this imbalanced scenario since it directly leverages the Teacher's text assignments, giving it a strong initialization when the vast majority of unlabeled samples come from unseen categories. This advantage disappears once the distribution becomes less extreme, as represented by the performance on the 20% New settings and the standard performance in the 50/50 split from Table 1 in the main paper. We acknowledge this as a limitation in the extremely unbalanced setting, and we have integrated this discussion in the main paper (**Revised Limitations Section**). These new results are provided in the **Appendix H, Table 19**.
>
> ## On overfitting to Old classes (W6)
>
> In the context of GCD, "overfitting to Old classes" does not necessarily manifest as lower Old accuracy, but rather as an imbalance: the classifier tends to allocate excessive probability to Old categories because they receive labeled supervision, causing many New samples to be absorbed into Old prototypes. This results in a large Old–New performance gap, even if Old accuracy remains high.
>
> Cross-modal representations mitigate this bias by anchoring the classifier to a broader semantic space. This often produces a more balanced allocation between Old and New classes. As a consequence, a small decrease in Old accuracy may occur, but it is accompanied by a substantial improvement in New accuracy, as shown in Figure 3, Figure 7 from the Appendix and Table 9 also from the Appendix.
>
> This behavior reflects a common trade-off in GCD: slightly less confidence on Old categories leads to significantly better discrimination of unseen ones. Importantly, the overall ALL accuracy still improves, confirming that the cross-modal variant provides better generalization despite a modest reduction in Old accuracy.

---

> ### Author Response · Authors · 2025-11-22
> **Responses to Reviewer hK7S (3/3)**
>
> ## On incremental Spectral Filtering (W7)
>
> Spectral Filtering is performed only once per dataset and is used to produce a fixed filtered dictionary $\hat{\mathcal{C}}$. When new unlabeled data arrive, recomputing the filtering step is inexpensive: it requires a single forward pass of the frozen teacher plus an eigendecomposition of the covariance matrix. As reported in Figure 4 and described in the section Training Efficiency, performing this on the whole dataset takes 194s on CUB, an order of magnitude faster than GET (3121s) and comparable to the TextGCD preparation phase (102s + LLM preprocessing).
>
> Moreover, we argue that the filtered dictionary need not be recomputed unless the incoming data has a substantial domain shift. To test this, we perform additional experiments in which Spectral Filtering is computed only from Old samples (either all Old or only labeled Old). As shown below, performance remains close to the full version:
>
> ### Spectral Filtering on different subsets (CUB)
>
> |Filtering Data Split|CUB ALL|CUB OLD|CUB NEW|
> |-|-|-|-|
> |Full Dataset|**79.2**|80.4|**78.5**|
> |Only Old (Labeled + Unlabeled)|77.8|**80.8**|76.4|
> |Only Old (Labeled)|78.0|80.4|76.8|
> | |
>
> These results indicate that SpectralGCD is robust to moderate distribution changes, since it works with a dictionary filtered using only partial knowledge of the full distribution of the dataset. Even so, if the distribution changes significantly and recomputing the full filtering step becomes the best alternative, the overall pipeline remains significantly cheaper than other multimodal GCD methods (see Figure 9, Appendix F). We added this experiment in the revised version (**Table 20, Appendix I**).
> Finally, we agree that designing a fully continual or online variant of Spectral Filtering is an interesting direction for future work, but our current method is not specifically designed for continual learning.
>
> ## On reverse distillation (W8)
>
> Thank you for this interesting question. We argue that instead of the reverse distillation being a problem when the teacher is wrong, the forward is more problematic in this situation since it forces the student to match the probability of the highest probability concepts in the teacher.  This intuition is empirically shown in Table 2, Table 11 of our paper, where we show that forward distillation results in higher Spearman correlation compared to reverse distillation, but this does not equal a better performance in terms of accuracy.
>
> We also note that, in some cases, using only the reverse term actually helps in achieving better performance compared to also using both directions, as shown in Table 10 on the CUB dataset. This is because while forward distillation forces the student to match high probability concepts of the teacher, the reverse actually does the opposite: it tells where *not* to focus and reduce the probability of the highly unlikely concepts.
>
> To better show this we performed additional distillation ablations on the Aircraft dataset, where our teacher CLIP ViT-H/14 achieves underwhelming zero-shot performance (obtained by assuming access to both OLD and NEW class names using standard CLIP-style Zero-Shot inference).
>
> |Method|Aircraft ALL|Aircraft OLD|Aircraft NEW|
> |-|-|-|-|
> |CLIP H/14 Zero-Shot|43.2|39.1|45.2|
> |**Ours** (FD)|59.7|60.1|59.5|
> |**Ours** (RD)|61.4|63.8|60.2|
> |**Ours** (FD + RD)|**63.0**|**66.1**|**61.4**|
> | |
>
> We note that our method performing only reverse distillation (RD) outperforms the only forward (FD) one. Moreover, we are able to improve over the teacher Zero-Shot performance, showing that we are not strictly bounded by the teacher knowledge.
>
> The same pattern appears on the New Energy Vehicle categories (NEV) dataset, previously introduced by GET [1]: reverse distillation outperforms forward distillation and even the combined variant in this particular dataset, similar to what we observed on CUB in Table 10 in our paper.
>
> |Method|NEV ALL|NEV OLD|NEV NEW|
> |-|-|-|-|
> |Zero-Shot (H14 backbone)|40.4|40.6|40.3|
> |SimGCD (CLIP backbone)|78.6$\pm$ 7.3|95.6$\pm$ 2.2|70.1$\pm$ 10.2|
> |GET|83.2$\pm$ 2.6|98.5$\pm$ 0.9|75.5$\pm$ 3.8|
> |**Ours** (No KD)|79.8$\pm$ 6.9|97.6$\pm$ 2.3|70.9$\pm$ 10.5|
> |**Ours** (FD)|81.9$\pm$ 5.1|**99.1**$\pm$ 0.4|73.3$\pm$ 7.5|
> |**Ours** (RD)|**84.4**$\pm$ 1.3|97.3$\pm$ 1.7|**77.9**$\pm$ 1.8|
> |**Ours** (FD+RD)|83.3$\pm$ 1.4|97.7$\pm$ 1.0|76.1$\pm$ 2.1|
> | |
>
> Overall, using both distillations helps in achieving a better trade-off of both higher Spearman correlation and higher accuracy, as shown by the results on our paper and the additional results we reported here. We have integrated these additional results on NEV in **Appendix E.1, Table 15** of the revised manuscript.
>
> [1] GET: Unlocking the multi-modal potential of clip for generalized category discovery. CVPR 2025.

---

### Official Review · Reviewer_GDAc · 2025-11-01

**Soundness:** 3
**Presentation:** 3
**Contribution:** 2
**Rating:** 4
**Confidence:** 5

**Summary:**

The paper addresses Generalized Category Discovery, aiming to find novel categories using limited labeled data. It introduces SpectralGCD, which builds a unified cross-modal representation by expressing images as mixtures over CLIP-derived semantic concepts. A teacher-guided Spectral Filtering selects relevant concepts via a cross-modal covariance matrix, and bidirectional knowledge distillation keeps the student’s representations aligned and semantically sufficient. On six benchmarks, SpectralGCD matches or surpasses SOTA.

**Strengths:**

(1) The idea of using the cross-modal representations is interesting.

(2) The paper is clearly written and easy to follow.

(3) The performance is promising.

**Weaknesses:**

(1) Using VLMs (e.g., CLIP) for GCD risks data leakage, as these models may have been exposed to images or names of the “unknown” classes. Prior work (e.g., GET) evaluates on splits unseen by CLIP to mitigate this. Please discuss this issue and, if possible, include experiments on CLIP-unseen splits or provide a robustness analysis addressing the leakage problem.

(2) What is the performance when using ViT-B/16 as the teacher or using ViT-H/14 as student? To what extent do the gains stem from distillation from a larger teacher rather than the proposed components? An ablation varying teacher and student capacity (e.g., ViT-B vs ViT-H) would help isolate the contribution.

(3) Please report or elaborate on the zero-shot performance of the CLIP models in Table 1, to contextualize the improvements over zero-shot.

(4) The KD component seems fairly standard and lacks technical novelty. Please clarify the insight beyond common KD practices.

(5) What are the total inference costs compared with multimodal methods (GET, TextGCD) and unimodal baselines (SimGCD)? Latency would clarify efficiency trade-offs, as the proposed approach appears quite complex.

(6) Have you evaluated fine-tuning the text encoder? Reporting this result would be informative.

(7) The abstract states: “Training a parametric classifier solely on image features often leads to overfitting to old classes.” Is this primarily due to the absence of labeled images for the novel classes during training, which biases the classifier toward seen (old) categories? Please clarify more about this issue.

**Questions:**

See Weaknesses.

---

> ### Author Response · Authors · 2025-11-22
> **Responses to Reviewer GDAc (1/3)**
>
> We thank the reviewer for their review and for recognizing the novelty of our cross-modal representation, for noting that the performance of our approach surpasses or is comparable to the current state-of-the-art, and for appreciating the clarity of our manuscript. Below we respond to specific points made in the initial review.
>
> ## On data leakage when using CLIP (W1)
>
> This is an interesting question and we have performed additional experiments on the NEV dataset introduced by GET [1] with New Energy Vehicle categories introduced in 2023 and not included in CLIP pre-training, which was also included as an ablation in the GET paper. We also reproduced SimGCD and GET on this dataset, performing five runs on different seeds due to the high variance demonstrated by all approaches:
>
> |Method|NEV ALL|NEV OLD|NEV NEW|
> |-|-|-|-|
> |Zero-Shot (H14 backbone)|40.4|40.6|40.3|
> |SimGCD (CLIP backbone)|78.6 $\pm$ 7.3|95.6 $\pm$ 2.2|70.1 $\pm$ 10.2|
> |GET|83.2 $\pm$ 2.6|98.5 $\pm$ 0.9|75.5 $\pm$ 3.8|
> |**Ours** (No KD)|79.8 $\pm$ 6.9|97.6 $\pm$ 2.3|70.9 $\pm$ 10.5|
> |**Ours** (FD)|81.9 $\pm$ 5.1|**99.1** $\pm$ 0.4|73.3 $\pm$ 7.5|
> |**Ours** (RD)|**84.4** $\pm$ 1.3|97.3 $\pm$ 1.7|**77.9** $\pm$ 1.8|
> |**Ours** (FD+RD)|83.3 $\pm$ 1.4|97.7 $\pm$ 1.0|76.1 $\pm$ 2.1|
>
> We see that even on categories to which CLIP was never before exposed, leveraging the text modality allow both SpectralGCD and GET to improve upon unimodal techniques. We added these results in **Table 15 (Appendix E.1)**.
>
> To perform these experiments, we set the distillation weight $\lambda_{\text{kd}} = 0.5$ (instead of 1.0 as on other datasets) in SpectralGCD. Because NEV includes object categories that are not present in CLIP’s pre-training data, the teacher is naturally less reliable on this distribution. In such cases, a strong distillation signal hinders student learning and lowering the weight gives more stable and reliable results, while still avoiding semantic drift.
>
> We report the ablation of $\lambda_{\text{kd}}$ for the CUB and StanfordCars datasets evaluated in the main paper in **Table 6** of the revised paper. For the novel NEV dataset we provide the corresponding ablation in **Appendix E.1, Table 16**.
>
> [1] GET: Unlocking the multi-modal potential of clip for generalized category discovery. CVPR, 2025
>
> ## On teacher and student capacity (W2)
>
> An ablation comparing different teacher sizes is given in Tables 4 and Table 14 in the original submission. In them we evaluate as teachers: ViT-B/16, ViT-H/14 (LAION-2B), and ViT-H/14-QuickGELU (DFN-5B). Results show that a stronger teacher transfers higher-quality representations to the student, corresponding to higher performance.
>
> We note, however, that the performance of SpectralGCD is not bounded by the capabilities of the teacher, as in Table 1 we see that SpectralGCD is able to *improve* over teacher zero-shot performance (which assumes knowledge of both Old and New class names) on multiple benchmarks.
>
> To quantify the impact of different student architectures, we have run additional experiments on the **CUB** dataset by matching the student backbone to the teacher's (i.e. ViT-H/14, LAION-2B). For a fair comparison, we report the original SimGCD and TextGCD results with the same architecture from [2]:
>
> |Method (Backbone)|CUB ALL|CUB OLD|CUB NEW|
> |-|-|-|-|
> |TextGCD (ViT-B/16)|76.6|80.6|74.7|
> |**Ours** (ViT-B/16)|79.2|80.4|78.5|
> |SimGCD (ViT-H/14)*|69.1|76.3|65.4|
> |TextGCD (ViT-H/14)*|78.6|81.5|77.1|
> |**Ours** (ViT-H/14)|**80.0**|**82.9**|**78.6**|
>
> Increasing student capacity leads to a small but consistent improvement in overall accuracy, indicating that our method can benefit from a stronger student backbone. The gains, however, are modest and the performance on NEW classes is essentially unchanged. Crucially, even when using the same high-capacity architecture as the teacher (ViT-H/14), our approach still surpasses SimGCD and TextGCD that also use H/14. This suggests that the improvements are primarily driven by the contributions of SpectralGCD rather than by model scaling alone. We added these new results in the revised main paper in **Section 5.3, Table 5**.
>
> [2] Textual knowledge matters: Cross-modality co-teaching for generalized visual class discovery. ECCV, 2024.

---

> ### Author Response · Authors · 2025-11-22
> **Responses to Reviewer GDAc (2/3)**
>
> ## On comparison with zero-shot CLIP performance (W3)
>
> Note that Table 1 already reported the zero-shot performance of both the CLIP ViT-B/16 (student) and CLIP ViT-H/14 (teacher) models. These results were obtained by assuming access to both OLD and NEW class names and using the classic CLIP-style zero-shot inference. We additionally discuss these results in the main text (Section 5.2, lines 419-426). These comparisons contextualize the improvements over zero-shot CLIP and show that SpectralGCD does not simply inherit the teacher's accuracy but rather learns a stronger cross-modal representation that guides the classification and discovery process more robustly. We highlight this more explicitly in **Table 1** in the revised version.
>
> ## On the novelty of Knowledge Distillation (W4)
>
> We do not claim knowledge distillation as a novel contribution. The novelty lies in how and why we apply it in the cross-modal GCD setting. Instead of distilling class predictions, we perform bi-directional distillation over concept-similarity scores (which we call *cross-modal representations*) produced by a frozen vision–language teacher. This is crucial for stabilizing the student’s cross-modal representation during parametric training and for preventing semantic drift, as shown in Tables 2 and 10 in our paper.
>
> To best of our knowledge, this is the first work employing this distillation strategy in GCD. Our ablations (Tables 2 and 10) show that both directions of distillation significantly improve robustness and representation quality.
> Finally, we note that it precisely our novel cross-modal representation that renders knowledge distillation possible. By mapping into the space of image-concept similarities, we are able to perform knowledge distillation using *any* model capable of producing image-concept similarity scores.
>
> ## On total inference costs (W5)
>
> Our method is designed to remain as efficient as unimodal GCD methods at inference time: unlike TextGCD and GET, SpectralGCD performs no text encoder forward passes at inference and uses only the student image encoder, one linear projection, and one classifier, matching the low computational cost of SimGCD. Moreover, the Spectral Filtering phase is a one-time offline preprocessing step and does not affect inference at all. For clarity, we report inference times on CUB and CIFAR100.
>
> ### CUB:
> |Method|Inference Time (s) ↓|
> |-|-|
> |SimGCD|4.20|
> |**Ours**|4.23|
> |GET|6.59|
> |TextGCD|8.95 (+12.25 for text assignments)|
> | |
>
> ### CIFAR100:
> |Method|Inference Time (s) ↓|
> |-|-|
> |SimGCD|21.20|
> |**Ours**|21.30|
> |GET|37.12|
> |TextGCD|52.69 (+16.40 for text assignments)|
> | |
>
> We also note that, if the test set to evaluate on has never been seen before, TextGCD must precompute text assignments using the teacher model, significantly increasing overall inference costs on new data. This additional overhead required for TextGCD is also reported in the table above. We added these results in **Appendix F, Table 17**.

---

> ### Author Response · Authors · 2025-11-22
> **Responses to Reviewer GDAc (3/3)**
>
> ## On fine-tuning the text encoder (W6)
>
> We intentionally keep the text encoder frozen because the text embeddings define a stable semantic basis for measuring image–concept similarities. Fine-tuning both encoders causes them to drift together, breaking this anchor. To prove the efficacy of our design-choice we have compared our parameter efficient *Vision only* fine-tuning with the addition of full text-encoder fine-tuning (*Vision + Text*) and we report the results below.
>
> |Method|CUB ALL|CUB OLD|CUB NEW|Cars ALL|Cars OLD|Cars NEW|
> |-|-|-|-|-|-|-|
> |Vision only|**79.2**|80.4|**78.5**|**89.1**|**92.6**|**87.4**|
> |Vision + Text|77.6|**81.0**|75.9|87.7|91.3|86.0|
> | |
>
> While OLD categories see marginal gains on CUB, this comes at the expense of a degradation in discovering NEW categories. This behavior aligns with recent findings on vision-language model adaptation [3,4], which demonstrate that keeping the text encoder frozen preserves the pre-trained semantic structure essential for generalization. In our setting, the frozen text space provides the semantic foundation for spectral filtering, enabling discovery of novel categories through their relationship to known concepts. Fine-tuning causes the text encoder to overfit to labeled OLD categories, losing the general semantic structure needed in GCD. We added these results in **Appendix G, Table 18**.
>
> [3] CLIPood: Generalizing clip to out-of-distributions. ICML 2023.
>
> [4] CLIP-Adapter: Better Vision-Language Models with Feature Adapters. IJCV 2022.
>
> ## On overfitting to Old classes (W7)
>
> This is a well-established issue in GCD, and we cite several works that document it (e.g., [5], [6]). As noted in the paper (in Introduction lines 44-45 and Sec. 4.1 lines 216-221), the parametric classifier receives supervision only from Old classes, and with few labeled samples it tends to overfit to spurious visual cues specific to Old classes. At the same time, the unsupervised branch provides noisy signals for New classes, causing the model to systematically bias predictions toward Old categories. This is also showed by how empirical results from Figure 3, Figure 6, Figure 7, and Table 9.
>
> [5] DebGCD: Debiased Learning wth Distribution Guidance for Generalized Category Discovery. ICLR 2025.
>
> [6] MOS: Modeling Object-Scene Associations in Generalized Category Discovery. CVPR 2025.

---

> > ### Comment · Reviewer_GDAc · 2025-11-26
> > **Response to authors' rebuttal**
> >
> > Thank you for the authors’ rebuttal. Some of my concerns have been addressed, but I have a few follow-up questions:
> >
> > (1) I have reservations about some of the experimental results. In Table 1 there is a large, expected zero‑shot performance gap between the ViT-B/16 and ViT-H/14 CLIP models, yet in the table provided in the rebuttal to W2 the difference between using ViT-H/14 or ViT-B/16 as teacher is marginal. Moreover, as a partially labelled task, GCD methods can show large run‑to‑run variance. The results on CUB can not give a definite conclusion. Experiments on additional datasets (including generic datasets) and reporting of variance (e.g., multiple runs with mean ± std) are essential to resolve this concern.
> >
> > (2) Regarding time cost: what is the total training time (including the spectral filtering step) compared to other methods?
> >
> > (3) The justification for “Spectral Filtering” and the novelty of the proposed method remain debatable, as also noted by reviewers YwSE and hK7S.

---

> > > ### Author Response · Authors · 2025-11-26
> > > **Teacher/student ablations and training times**
> > >
> > > ## (1) On experiments varying student and teacher backbones
> > >
> > > **Varying the Student Backbone**. It appears the previous table may have created a misunderstanding of our intended message. The table provided in the response to W2 above does not compare different *teacher* models; rather, it **varies the student backbone**, as requested by the reviewer in the original review. To avoid ambiguity, we have now improved the table formatting to clarify the teacher/student columns. Moreover, we report mean and standard deviation over three runs, and have added CIFAR100 (a general object recognition dataset) as requested. Methods marked with * correspond to SimGCD and TextGCD results taken from the TextGCD paper which does not provide standard deviations:
> > >
> > > |Method|Student Model|Teacher Model|CUB ALL|CUB OLD|CUB NEW|CIFAR100 ALL|CIFAR100 OLD|CIFAR100 NEW|
> > > |-|-|-|-|-|-|-|-|-|
> > > |TextGCD|ViT-B/16|ViT-H/14|$76.6 \pm 1.0$|$80.6 \pm 0.8$|$74.7 \pm 1.1$|$85.7 \pm 0.6$|$86.3 \pm 0.6$|$\textbf{84.6} \pm 0.9$|
> > > |**Ours**|ViT-B/16|ViT-H/14|$79.2 \pm 0.9$|$80.4 \pm 0.4$|$78.5 \pm 1.1$|$86.1 \pm 0.8$|$87.2 \pm 0.3$|$83.9 \pm 2.3$|
> > > |SimGCD*|ViT-H/14  |-|$69.1$|$76.3$|$65.4$|$78.1$|$80.0$|$74.4$|
> > > |TextGCD*|ViT-H/14 |ViT-H/14|$78.6$|$81.5$|$77.1$|$\textbf{86.4}$|$\textbf{89.3}$|$80.7$|
> > > |**Ours**| ViT-H/14)|ViT-H/14|$\textbf{80.0} \pm 0.6$|$\textbf{82.9} \pm 0.3$|$\textbf{78.6} \pm 1.0$|$85.7 \pm 0.4$|$88.9 \pm 0.7$|$79.4 \pm 0.4$|
> > > ||
> > >
> > > The reviewer is correct in that there are marginal differences in terms of performance when changing the student, which is due to the fact that all models were trained using the same teacher. Interestingly, using a stronger student backbone does not yield substantially better results than the smaller ViT-B/16 when using SpectralGCD, as knowledge distillation via the cross-modal representation is able to effectively transfer knowledge from student to teacher.
> > >
> > > **Varying the Teacher Backbone**. In our [previous response to W2](https://openreview.net/forum?id=PyfV9tFmdR&noteId=gWBJQBGW3A) we attempted to clarify that this experiment was already included in the original submission (Table 4 in the main paper) and the reliance on the teacher discussed in the Limitations Section. For convenience, we report that table here, now including the standard deviations over three runs as requested by the reviewer:
> > >
> > > Student Model |Teacher Model|CUB ALL|CUB OLD|CUB NEW|CIFAR100 ALL|CIFAR100 OLD|CIFAR100 NEW|
> > > |- |-|-|-|-|-|-|-|
> > > ViT-B/16|ViT-B/16|$72.7 \pm 0.6$|$74.2 \pm 0.7$|$71.9 \pm 0.7$|$78.5 \pm 1.2$|$82.5 \pm 0.7$|$70.4 \pm 2.8$|
> > > ViT-B/16|H/14 (LAION-2B)|$79.2 \pm 0.9$|$80.4 \pm 0.4$|$78.5 \pm 1.1$|$86.1 \pm 0.8$|$87.2 \pm 0.3$|$83.9 \pm 2.3$|
> > > ViT-B/16|H/14-QuickGelu (DFN-5B)|$80.6 \pm 0.8$|$81.8 \pm 0.3$|$80.0 \pm 1.3$|$88.8 \pm 1.9$|$90.9 \pm 0.1$|$84.6 \pm 5.8$|
> > > ||
> > >
> > > ViT-H/14 (LAION-2B) is the same teacher used in TextGCD. ViT-H/14-QuickGELU (DFN-5B) is a stronger teacher that we introduce only for this experiment, since it has a larger and different pre-training dataset, to show the impact of using a more powerful teacher. In Table 14 we provide additional results for Stanford Cars.
> > >
> > >
> > > ## (2) On total training times
> > >
> > > We reported the total training times including all preprocessing steps in the original submission (see **Figure 4, Section 5.2** in the **main paper**). To complement those results, in the original submission we also provided in **Figure 9, Appendix F** the training times for all the methods across all the datasets. *All reported timings include the spectral filtering step, and similarly for TextGCD and GET include their pre-processing phases*.
> > >
> > > For comparison, on ImageNet-100, SimGCD requires about 10.5 hours, SpectralGCD (Our) requires approximately 11 hours, GET around 36 hours, and TextGCD roughly 45.5 hours. SpectralGCD requires a total training time comparable to the unimodal SimCGD method, while the other multimodal GCD techniques (GET and TextGCD) require significantly more.

---

> > > ### Author Response · Authors · 2025-11-26
> > > **Spectral Filtering and the novelty of our contributions**
> > >
> > > ## (3) On the justification of Spectral Filtering
> > >
> > > Our goal in introducing Spectral Filtering, and the entire pipeline of SpectralGCD, is to address the inherent difficulty of leveraging the semantic information of CLIP without having explicit access to any knowledge of the novel classes. To our knowledge, no other multimodal GCD method has filtered and exploited such a cross-modal representation as we propose in this work. TextGCD uses an LLM to generate rich descriptions, starting from the same large concept dictionary we use and then uses the teacher model to hard-assign one caption to each sample, while GET trains an inversion network to produce a pseudo-text embedding. Both strategies are fundamentally different from ours, although they share our same goal of deriving discriminative, semantic information about new, unlabeled data.
> > >
> > > Regarding the justification of Spectral Filtering, we feel we have amply demonstrated by both empirical results (which we provided in our experimental results) and by the new qualitative and quantitative analyses provided in our responses to [Reviewer YwSE](https://openreview.net/forum?id=PyfV9tFmdR&noteId=jz8KaN2lEN). Spectral Filtering is a simple yet extremely effective approach to identifying salient concepts -- and discarding non-salient features -- from a large task-agnostic dictionary. The cross-modal representation that results from it is discriminative, as demonstrated by the linear probing and image retrieval experiments reported in our response to [Reviewer YwSE](https://openreview.net/forum?id=PyfV9tFmdR&noteId=jz8KaN2lEN) and included in the revised manuscript.
> > >
> > > We emphasize that *a key contribution of our work is the proposed cross-modal representation*. By focusing on image-concept similarities, this representation is independent of both the multimodal encoder and the downstream GCD architecture. This flexibility is the feature that makes knowledge distillation possible, not only between the teachers and students considered in our work, but *from any model able to generate concept similarity scores*. SpectralGCD integrates visual and textual semantics early on, eliminating the requirement for separate text and image classifiers as in TextGCD. Coupled with our approach for identifying the most relevant concepts, we firmly believe that SpectralGCD marks a significant and practical advancement in the multimodal GCD state-of-the-art.
> > >
> > > We thank the reviewer for engaging with our work and initiating this discussion, and we hope that with these clarifications we have addressed any concerns the reviewer has regarding our experimental results or the justifications behind the SpectralGCD approach.

---

> ### Comment · Reviewer_GDAc · 2025-11-27
> **Response to the authors' rebuttal**
>
> Thank you for the authors’ response. Based on the reported results, the observed performance gains appear to come mainly from using the ViT-H/14 teacher model, which is substantially larger and trained on much more data. When the smaller ViT-B/16 is used as the teacher, performance is even lower than the SimGCD baseline reported in [A], which calls the method’s advantages into question. Moreover, because the use of VL models and distillation from larger VL models has already been explored in the GCD literature[A,B], I believe the paper’s overall contribution is limited and requires further work to meet the standards of a top-tier venue such as ICLR. Therefore, I will maintain my original score.
>
> [A] GET: Unlocking the Multi-Modal Potential of CLIP for Generalized Category Discovery. CVPR 2025.
>
> [B] Textual Knowledge Matters: Cross-Modality Co-Teaching for Generalized Visual Class Discovery. ECCV 2024.

---

> > ### Author Response · Authors · 2025-11-27
> > **Our position regarding Reviewer claims**
> >
> > We appreciate the Reviewer for their continued engagement with our work. We would like to clarify some points in this response, as there may be some fundamental misunderstandings regarding our experiments and the intersection between our work and the relevant GCD literature. To facilitate a clearer discussion, we will address each point inline.
> >
> > > Thank you for the authors’ response. Based on the reported results, the observed performance gains appear to come mainly from using the ViT-H/14 teacher model, which is substantially larger and trained on much more data. When the smaller ViT-B/16 is used as the teacher, performance is even lower than the SimGCD baseline reported in [A], which calls the method’s advantages into question.
> >
> > All of our main experimental results *deliberately* use a *small* ViT-B/16 backbone as *student* and a *stronger* ViT-H14 backbone as teacher. This is the fundamental point of knowledge distillation: to transfer knowledge from a powerful teacher to a smaller, more efficient student. The situation the reviewer refers to is a single, isolated ablation in which -- when deliberately using a *weak* ViT-B/16 teacher in our method for the purposes of **ablation** -- SimGCD **marginally** surpasses this weakened variant **only on CIFAR-100**. Note that TextGCD uses *exactly* the same ViT-H/14 backbone as us for hard assignment of tags to images.
> >
> > For convenience, we repeat our ablations -- all of which were included in the **original manuscript**, as emphasized in our previous response -- using the weak ViT-B/16 backbone as teacher on all datasets:
> >
> > |Method|Student Model |Teacher Model|CUB ALL|CUB OLD|CUB NEW|Cars ALL|Cars OLD|Cars NEW|CIFAR100 ALL|CIFAR100 OLD|CIFAR100 NEW|
> > |-|-|-|-|-|-|-|-|-|-|-|-|
> > |SimGCD (reproduced by GET)|CLIP ViT-B/16|-|$71.7$|$\textbf{76.5}$|$69.4$|$70.0$|$83.4$|$63.5$|$\textbf{81.1}$|$\textbf{85.0}$|$\textbf{73.3}$|
> > |Ours|CLIP ViT-B/16|CLIP ViT-B/16|$\textbf{72.7} \pm 0.6$|$74.2 \pm 0.7$|$\textbf{71.9} \pm 0.7$|$\textbf{75.1} \pm 0.4$|$\textbf{84.7} \pm 0.8$|$\textbf{70.4} \pm 0.5$|$78.5 \pm 1.2$|$82.5 \pm 0.7$|$70.4 \pm 2.8$|
> > ||
> >
> > While true that our results using ViT-B/16 as both teacher and student (i.e. *self-distillation*) are slightly inferior on *one* of these datasets, SpectralGCD *still outperforms SimGCD on all others*. This demonstrates that, even in this weakened configuration, our gains do not stem from relying on a larger pretrained model, but from the design of our approach itself, namely the benefits of introducing a unified cross-modal representation, the Spectral Filtering mechanism, and the use of bidirectional distillation to maintain semantic alignment.
> >
> > To remove any doubts regarding dependence on teacher strength, we also compared all approaches using the *same ViT-H/14 backbone* -- as requested by the reviewer and reported above in our previous response. In this setting, SpectralGCD shows clear improvements over SimGCD on both CUB (+10.9%) and CIFAR-100 (+7.6%), and performance that improves over TextGCD on CUB while remaining comparable on CIFAR-100. This demonstrates again the benefits of our approach.
> >
> > >
> > > Moreover, because the use of VL models and distillation from larger VL models has already been explored in the GCD literature[A,B], I believe the paper’s overall contribution is limited and requires further work to meet the standards of a top-tier venue such as ICLR. Therefore, I will maintain my original score.
> >
> > This statement is, quite simply, *false*. **Neither GET [A] nor TextGCD [B] employ any form of knowledge distillation in their approaches**. GET performs no knowledge distillation, but trains an **inversion network** to map images into the textual space and then trains a shared, parametric classifier. TextGCD uses the teacher CLIP model ViT-H/14 **only for caption assignment** and not for any type of distillation. It then trains two modality-specific classifiers, again **without any distillation mechanism**.
> >
> > None of the key components of our method, not our **cross-modal representation**, not our proposed **Spectral Filtering**, and not our **use of distillation to regularize cross-modal semantics** have been explored in any prior GCD or multimodal GCD work.
> >
> > In light of these fundamental misunderstandings, we respectfully invite the Reviewer to reconsider their position and reevaluate their opinion regarding our contributions.
> >
> > [A] GET: Unlocking the Multi-Modal Potential of CLIP for Generalized Category Discovery. CVPR 2025.
> >
> > [B] Textual Knowledge Matters: Cross-Modality Co-Teaching for Generalized Visual Class Discovery. ECCV 2024.

---

> > > ### Comment · Reviewer_GDAc · 2025-11-28
> > > **Response to the authors' rebuttal**
> > >
> > > Thank you to the authors for their response and the substantial effort in the rebuttal. After further consideration and re-evaluation, I agree that some techniques are novel to the area; however, leveraging VL models and the introduction of larger VL models (although it's not direct output KD) has been explored in the GCD literature. Besides, I remain concerned about the limited improvement over the SimGCD baseline when a ViT-H teacher is not used. I will raise my score to 6 and leave further assessment to the other reviewers and the AC.

---

### Official Review · Reviewer_YwSE · 2025-11-02

**Soundness:** 2
**Presentation:** 3
**Contribution:** 2
**Rating:** 4
**Confidence:** 5

**Summary:**

This paper tackles the problem of Generalized Category Discovery (GCD), where models often overfit to known classes. The authors propose SpectralGCD, a multimodal approach that represents images not by their visual features directly, but as a mixture over semantic concepts from a large dictionary. This "cross-modal representation" is derived from CLIP-based image-concept similarities. The core technical contribution is "Spectral Filtering," a method that automatically prunes the concept dictionary by performing an eigendecomposition on a cross-modal covariance matrix derived from a strong teacher model, retaining only concepts deemed most informative. The learning process for a smaller student model is then guided by a combination of standard GCD losses and a forward-and-reverse knowledge distillation objective to align its representations with the teacher's.

**Strengths:**

1. The paper addresses the Generalized Category Discovery  task, focusing on the common problem where models overfit to the labeled "Old" classes and perform poorly on unlabeled "New" classes.

2. Proposed Core Idea: It introduces a novel "cross-modal representation" for each image. Instead of using raw image features, it represents an image as a vector of similarity scores against a large, task-agnostic dictionary of semantic concepts, computed using a pre-trained CLIP model.

3. To refine this representation and reduce noise from irrelevant concepts, the paper proposes "Spectral Filtering." This technique uses a strong teacher model to compute a cross-modal covariance matrix across the entire dataset. Through eigendecomposition (PCA), it identifies and retains concepts that contribute most to the principal components (i.e., high-variance directions) of the concept-similarity space.

**Weaknesses:**

1. The primary weakness lies in the justification for "Spectral Filtering". The motivation is to select "task-relevant" concepts. However, the mechanism (performing PCA on the global cross-modal covariance matrix) selects concepts that explain the most variance across the dataset. High variance does not necessarily equate to high discriminative power or task relevance. For example, a common background (e.g., 'sky', 'grass') present across many different classes could easily form a principal component with high variance. The method might then prioritize these non-discriminative concepts.

2. The paper makes a conceptual leap by equating "high contribution to dataset variance" with "semantic relevance for classification" without providing a strong theoretical or empirical argument to support this crucial link.

3. The paper feels more like a report on a successful engineering recipe than a deep scientific inquiry. This lack of insight limits the paper's contribution. An outstanding paper should not only present a method that works but also provide the understanding that allows the community to build upon its core ideas. SpectralGCD in its current form feels more like a well-tuned heuristic than a principled approach, making it less inspiring for future exploration.

4. In the task of discovering general categories, there has already been similar work [1] that decomposes objects into combinations of various attributes (textual or visual). I believe there needs to be more comparative discussion with the current work.

5. In addition, there has been progress in the discussion on the information represented by the covariance matrix of features in general category discovery. A comparison with those works [2,3] should be made.


[1] Dissecting Generalized Category Discovery: Multiplex Consensus under Self-Deconstruction. In ICCV, 2025.

[2] Generalized Category Discovery via Token Manifold Capacity Learning. In Arxiv, 2025.

[3] Continual Generalized Category Discovery: Learning and Forgetting from a Bayesian Perspective. In ICML, 2025.

**Questions:**

1. The core assumption of Spectral Filtering is that concepts contributing most to the variance of the cross-modal covariance matrix are the most "relevant". Could you provide a more rigorous justification for this? How does this method distinguish between concepts that are genuinely discriminative and those that are simply common or part of a shared background, which could also lead to high variance?

2. The paper frames the problem as representing an image as a "mixture over semantic concepts." This is an appealing analogy to topic modeling. However, the current implementation simply uses a linear projection on the similarity vector. Did you explore enforcing a probabilistic constraint (e.g., ensuring the representation is a valid probability distribution over concepts) to more closely follow this analogy?

---

> ### Author Response · Authors · 2025-11-22
> **Responses to Reviewer YwSE (1/3)**
>
> We thank the reviewer for their constructive criticism of our contribution. Below we respond to specific points raised in the initial review.
>
> ## On variance and discriminativeness of cross-modal representations (W1, W2, Q1)
>
> The reviewer asks if common or background concepts seen across multiple images, and thus are not discriminative, could have high variance and consequently be mistakenly selected as task-relevant. These concepts would be a problem if we would have applied Spectral Filtering directly on the Teacher cross-modal representations. However, we are applying softmax to teacher cross-modal representations *before* computing the cross-modal covariance matrix (see Eq. (6) of the main paper and the definition of $q_i$ just before).
>
> This softmax step plays a role analogous to term weighting in Latent Semantic Analysis (LSA): it amplifies higher values, increasing their overall variance, while downplaying common or weakly-informative background components (see the next response for more on links to LSA). Moreover, given the well-known object bias of CLIP caused by the contrastive loss with short captions [1, 2, 3], the highest Teacher similarity scores correspond to foreground objects, not to background or common parts. The softmax plus the object-bias of CLIP make Spectral Filtering simple and effective for removing noise and less salient information from the dictionary while retaining task-relevant concepts. We added this discussion to **Section 4.2**.
>
> To qualitatively show the effectiveness spectral filtering, we report the top-10 selected concepts on the CUB dataset (a dataset containg 200 fine-grained bird species) obtained when filtering the WordNet dictionary (~60K concepts) using the concept importance score defined in Eq (8):
> - **#1** tern, **#2** warbler, **#3** woodpecker, **#4** vesper sparrow, **#5** indigo bunting, **#6** vermillion flycatcher, **#7** redheaded woodpecker, **#8** rusty blackbird, **#9** evening grosbeak, **#10** whippoorwill
>
> To further assuage Reviewer doubts, we also report the ranks of selected concepts that we believe should reasonably be considered common, less-discriminative background concepts for the CUB dataset:
> - **#354** beak,...,**#9950** wing,...,**#9700** branch,...,**#21623** leg,...,**#23420** grass,...,**#25964** tree,...,**#29177** bush,...,**#36945** leaf,...,**#43612** sky,...,**#44501** cloud
>
> Note that Spectral Filtering on CUB with WordNet selects only around ~220 top-ranked concepts, thus the filtered concept dictionary keeps the discriminative concepts and automatically discards the less discriminative background ones.
>
> Finally, to provide more articulated, empirical evidence of the discriminative power of the selected concepts, we have run two additional experiments comparing image features to our filtered cross-modal representations computed using the frozen Teacher ViT-H/14 on CUB and Cars.
>
> ### Linear Probing
>
> We follow the standard linear probing evaluation protocol, keeping the encoder frozen and training a linear classifier on the full training split, supposing access to ALL labels:
>
> |Feature Type|CUB Accuracy|Cars Accuracy|
> |-|-|-|
> |Image Features|**88.5**|**95.1**|
> |**Ours** (Cross-modal representations)|87.6|94.9|
> |  |
>
> ### Image Retrieval
>
> In this second experiment we perform image-to-image retrieval considering images of the same class as positive examples:
>
> |Feature Type|CUB mAP|CUB R@1|Cars mAP|Cars R@1|
> |-|-|-|-|-|
> |Image Features|56.3|80.4|61.8|88.9|
> |**Ours** (Cross-modal representations)|**62.1**|**81.7**|**75.4**|**91.4**|
> |  |
>
> The linear probing results show that the cross-modal representation (even in a smaller representation space of 263 concepts for CUB and 377 for cars) are as linearly separable as the original image features (of embedding dimension 1024). More interestingly, cross-modal features are better calibrated for the harder, unsupervised task of image-to-image retrieval, improving mAP by nearly 6% on CUB and 14% on Cars over the image features.
>
> We have added both the WordNet analysis (**Table 21**) and the experiments demonstrating discriminativeness of the cross-modal representation (**Tables 22** and **23**) in **Appendix J**.
>
> [1] Two Effects, One Trigger: On the Modality Gap, Object Bias, and Information Imbalance in Contrastive Vision-Language Models. ICLR 2025
>
> [2] Common Data Properties Limit Object-Attribute Binding in CLIP. GCPR 2025.
>
> [3] What to align in multimodal contrastive learning?. ICLR 2025

---

> ### Author Response · Authors · 2025-11-22
> **Responses to Reviewer YwSE (2/3)**
>
> ## On related methods (W4, W5)
>
> We thank the reviewer for pointing us to these works. We did not include them initially because MTMC [5] (arXiv May 2025) and ConGCD [4] (arXiv mid-August 2025 and published in the ICCV proceedings near the ICLR deadline) appeared shortly before our submission. VB-CGCD [6] considers Continual GCD and therefore falls outside the scope of our static GCD setting. Nevertheless, its Mahalanobis-based classifier is conceptually appealing and could also benefit static GCD; integrating similar ideas is an interesting direction for future work. That said, all three papers provide valuable insights.
>
> Both ConGCD and MTMC are *unimodal*, vision-only methods based on DINO v1/v2 encoders and do not use textual information, unlike our CLIP-based approach. Thus, strict numerical comparisons are not directly meaningful due to different backbones, modalities, and pre-training data. For this reason, a conceptual comparison is more constructive than one based solely on absolute performance.
>
> MTMC improves representations by maximizing the class token nuclear norm, while ConGCD enriches them by decomposing images into visual primitives via an iterative procedure over a frozen vision encoder, where image primitives capture atomic components of a global image (e.g., wings, engines and nose for aircraft). Both methods enhance visual representations by increasing **intra class visual diversity**; their improved inter-class separation therefore emerges **indirectly** from this enriched intra-class structure.
>
> In contrast, SpectralGCD operates in **cross-modal CLIP space**, selecting discriminative semantic directions. This directly improves **inter-class** separation through textual semantics, without relying on intra-class visual enrichment. Our student model, trained to match the teacher cross-modal representation via KD, remains anchored to this discriminative structure and further refines it during GCD training (as shown by improvements over the zero-shot teacher in Table 1). We added the citations to and a discussion of MTMC and ConGCD in **Section 2** of the revised paper.
>
> [4] Dissecting Generalized Category Discovery: Multiplex Consensus under Self-Deconstruction. ICCV 2025.
>
> [5] Generalized Category Discovery via Token Manifold Capacity Learning. arXiv 2025.
>
> [6] Continual Generalized Category Discovery: Learning and Forgetting from a Bayesian Perspective. ICML 2025.
>
> ## On the Topic Model analogy (Q2)
>
> Although our goal was not to force strict comparison of our approach with topic models, we can leverage similarities with Latent Semantic Analysis (LSA) to shed light on our spectral filtering method. We note again that the SVD is performed on the covariance of the *softmaxed* concept frequencies and *not* directly on the cross-modal representations (see Eq. (6) and definition of $q_i$ just before). Thus, the eigenvectors $V$ of this matrix are exactly the right singular vectors of the concept-frequency matrix one would obtain by stacking the concept-frequency vectors $q_i$ of all images. So, we can interpret each eigenvector as a topic consisting of unscaled logits over the concept dictionary. Seen in this light, our concept importance score (Eq. (8)), which is the basis for concept selection in Eq. (9), is computed in a way not dissimilar to how *top terms* are identified in LSA Topic Models by analyzing which terms are assigned most probability mass over all topics.
>
> We did experiment with using the softmax-normalized cross-modal representations for training the parametric classifier in Phase 2, but found the performance to be unsatisfactory. The softmax squashes too many logits to near-zero and destroys discriminative information necessary for the GCD task. We believe the link with generative topic models, like LDA, is an interesting direction and plan to investigate this further in future work.

---

> ### Author Response · Authors · 2025-11-22
> **Responses to Reviewer YwSE (3/3)**
>
> ## On the significance of our contribution (W3)
>
> The reviewer expresses the feeling that our work is an engineering recipe that, albeit successful, lacks the insight of a serious scientific inquiry. We respectfully disagree with this assessment. The primary challenge of employing multimodal models to the Generalized Category Discovery problem is how to exploit their strongly grounded semantics in the absence of any semantic information regarding unseen and unlabeled New classes. SpectralGCD offers *precisely* such a solution, outperforms the current state-of-the-art across multiple benchmarks, is 3x-4x more efficient at train time compared to the existing multimodal state-of-the-art, and maintains the same inference-time efficiency as unimodal approaches.
>
> These benefits, which we believe are amply demonstrated in our experimental results, directly derive from the cross-modal representation and the filtering mechanism used to select salient concepts. The cross-modal representation efficiently bridges the gap between image and text semantics to ground the base representation used for the semi-supervised GCD task. It requires no inversion network to inject text semantics into the representation (as GET does), nor does it require per-image assignment of ad hoc captions or paired text and image classifiers to perform GCD (as TextGCD does).
>
> Our proposed spectral filtering approach identifies concepts from a task-agnostic dictionary most salient to the task. TextGCD accomplishes a similar feat using an LLM to generate richer text descriptions starting from the concept vocabulary, and then hard-assigning *one* caption to every image. GET, on the other hand, trains a textual inversion network to derive a pseudo-text representation used for downstream GCD.
>
> To conclude, we emphasize that the major contribution of our work is the cross-modal representation. It renders the representation independent of both the multimodal model used to compute it and the downstream architecture used for GCD. By mapping into the space of image-concept similarities, we are able to perform knowledge distillation using *any* model capable of producing image-concept similarity scores -- something that GET cannot claim since it relies on a textual inversion network strongly tied to the transformer architecture. The cross-modal representation is an early fusion of visual and textual semantics, allowing us to perform GCD on a single representation rather than training independent text and image classifiers and fusing them downstream as TextGCD does. Because of these advantages, we firmly believe that our cross-modal representation, along with our demonstration of how salient and discriminative features can be identified, represents a significant advance in the state-of-the-at that the community can build upon.

---

> > ### Comment · Reviewer_YwSE · 2025-11-25
> >
> > Thank you to the authors for the response.
> >
> > In fact, I would like to have a more in-depth discussion with the authors regarding the original motivation of GCD, specifically the motivation for the model to "autonomously" identify novel categories.
> >
> > If CLIP is used, it implies that the issue of open-set problems, which GCD focuses on, can actually be addressed with CLIP's zero-shot capabilities. Given the involvement of **text** and the artificial constraints on the range of text, does this not deviate from the task setup of "autonomously" and the **original intention** behind the GCD task?

---

> > > ### Author Response · Authors · 2025-11-26
> > > **On CLIP and autonomous category discovery**
> > >
> > > We thank the reviewer for engaging with us in the discussion. We agree that the role of autonomy in GCD deserves clarification, especially in the multimodal setting with the CLIP model.
> > >
> > > In our view, GCD cannot be fully addressed by solely exploiting CLIP's zero-shot capabilities. This is because the ground truth *new unknown class names* in multimodal GCD -- including in our approach -- are never explicitly provided to the text encoder. Such class names are a prerequisite to zero-shot inference. Moreover, even when the **zero-shot teacher model** is given the real class names for both old and new classes (**i.e. an ideal oracle setting**), its performance remain substantially below that of SpectralGCD using a small ViT-B/16 as student (Table 1 of the main paper, the rows labeled **ZS CLIP B/16** and **ZS CLIP H/14**). Specifically: SpectralGCD, compared to the zero-shot *teacher* equipped with an oracle, improves by 6.6% on ImageNet100, 19.8% on Aircraft, 2.6% on Cifar100, and 1.8% on Cifar-10. Notably, on the NEV dataset, unseen by CLIP, *SpectralGCD gains 40% over the zero-shot oracle performance* **(Table 15, Appendix E.1)**.
> > >
> > > Given these results, our view is that the real potential of multimodal GCD is not reducible to simply using CLIP's zero-shot predictions to classify old and new classes. Rather, to realize this potential it is crucial to understand how to exploit CLIP's cross-modal alignment to learn representations that enable category discovery and how to further refine them during training. All of this must be done **without access to the true class names**, fully consistent with the standard GCD assumptions, and is far from trivial. From our perspective, this is conceptually similar to how unimodal GCD methods, like SimGCD, SelEx, ConGCD, MTMC, and VB-CGD exploit strong, pre-trained DINO backbones as priors which already exhibit high class separability and then attempt to refine them during training.
> > >
> > > Prior multimodal GCD approaches introduce different strategies to leverage CLIP's textual modality, without having access to true class names, to move towards this goal. Both GET, via textual inversion, and TextGCD, by performing hard assignments of LLM-generated text to images, automatically derive textual (or pseudo-textual in the case of GET) information relevant to the unseen and unlabeled classes without any supervision.
> > >
> > > The same applies to SpectralGCD. Having access to the image data and the large **task-agnostic** dictionary, our approach is able to *automatically* select task-relevant concepts without any additional supervision. This strategy is effective since using the whole concept dictionary yields sub-optimal performance. We do not view this as an artificial constraint, but rather as a principled way to leverage the structured semantic knowledge embodied in CLIP’s pre-training. SpectralGCD then uses these retained semantic concepts during training to further refine the student’s cross-modal representation via teacher distillation and parametric classifier training, which strengthens the model’s ability to discriminate between old and new classes.
> > >
> > > We thank the reviewer again for prompting us to reflect on the deeper implications of using CLIP and the relationship between the unimodal and multimodal approaches to GCD. We will incorporate elements of this discussion in the final version of this paper, as we feel it helps contextualize our contributions with respect to the broader literature.

---

### Official Review · Reviewer_ei3h · 2025-11-03

**Soundness:** 3
**Presentation:** 3
**Contribution:** 3
**Rating:** 8
**Confidence:** 4

**Summary:**

The authors propose a new GCD method inspired by probablistic models.

They model images as mistures over semantic concepts in CLIP embedding space.

They use spectral filtering to filter out irrelevant concepts from a large task agnostic dictionary.

They achieve better accuracy on standard benchmarks than existing unimodal and multimodal approaches to GCD.

**Strengths:**

- The idea of spectral filtering on a dictionary of concepts is cute.
- The paper is quite complex. There are multiple stages to the proposed method. Yet the proposed method is still efficient.

**Weaknesses:**

- The authors claim that the dictionary of concepts is task agnostic. But this isn't really true right? There must be some overlap between the dictionary and the concepts of the target dataset. Otherwise it would not work.
- GCD is useful for discovering concepts in a dataset that do not fit neatly into the existing label set. However, I would argue that the new assumptions that the authors are introducing render the task of GCD meaningless.
  - In particular, the authors use a Teacher model CLIP H/14 that has been pretrained on all the concepts (both new and old) across all the benchmarks. So the "novel classes" can't really be considered novel.
  - It would be more impressive if the authors test their method on datasets that have a smaller conceptual overlap with LAION. e.g. bacteria species classification based on cell cultures.

**Questions:**

Minor:
- It may be helpful to clarify how "New" accuracy is defined for those not familiar with the literature.

---

> ### Author Response · Authors · 2025-11-22
> **Responses to Reviewer ei3h**
>
> We thank the reviewer for their comments and recognition of the efficiency of our approach and the significance of our results compared to the state-of-the art uni- and multi-modal GCD. We are especially pleased they found interesting our idea of spectral filtering over a dictionary of concepts.
>
> ## On the task-agnosticity of the concept dictionary (W1)
>
> Our use of *task-agnostic* refers to the construction of the dictionary, not a claim of semantic independence from visual benchmarks. The dictionary is built *once* from a large, general-purpose corpus and *does not depend on dataset-specific tailoring or hand-crafting of the concept set*. We conducted additional experiments using WordNet, a general linguistic ontology that includes many concepts without visual grounding:
>
> |Method (Dictionary)|CUB ALL|CUB OLD|CUB NEW|Cars ALL|Cars OLD|Cars NEW|
> |-|-|-|-|-|-|-|
> |GET (N/A)|77.0|78.1|76.4|78.5|86.8|74.5|
> |TextGCD (WordNet)|69.8|76.2|66.6|63.6|84.7|53.4|
> |TextGCD (Tags)|69.9|74.9|67.3|86.2|91.2|83.8|
> |**Ours (WordNet)**|77.1|79.9|75.7|83.0|**93.5**|78.0|
> |**Ours (Tags)**|**79.2**|**80.4**|**78.5**|**89.1**|92.6|**87.4**|
>
> SpectralGCD remains robust in this setting and significantly outperforms TextGCD, although both use WordNet. It remains competitive or outperforms GET. This indicates that SpectralGCD can integrate broad, non-vision-specific concepts, and does not rely on dataset-specific tailoring of the dictionary (although domain-aligned concepts naturally are beneficial). We added these results in **Table 13 (Appendix D)**.
>
> ## On evaluation using unseen data (W2)
>
> We follow the standard GCD formulation ([1],[2]) with no additional assumptions. New classes are defined as categories that:
> 1. do not appear in the labeled subset; and
> 2. whose identities or names are never revealed during training.
>
> This definition is independent of what the backbone has seen during pretraining. Even if CLIP's pretraining contains images related to New classes, it does not provide class names or grouping information. In fact, Table 1 shows that Teacher zero-shot accuracy (having oracle access to the all class names) is *below* SpectralGCD on multiple datasets. If CLIP had already "solved" these categories, the Teacher accuracy would match or exceed our results.
>
> To show that our performance does not solely rely on pre-training data, we report results on the New Energy Vehicle categories (NEV) dataset introduced in [1] to evaluate performance on categories to which CLIP was never exposed (it was introduced after the release of CLIP). We also reproduce results for SimGCD and GET, performing five runs on different seeds due to the high variance of all approaches:
>
> |Method|NEV ALL|NEV OLD|NEV NEW|
> |-|-|-|-|
> |Zero-Shot (H14 backbone)|40.4|40.6|40.3|
> |SimGCD (CLIP backbone)|78.6 $\pm$ 7.3|95.6 $\pm$ 2.2|70.1 $\pm$ 10.2|
> |GET|83.2 $\pm$ 2.6|98.5 $\pm$ 0.9|75.5 $\pm$ 3.8|
> |**Ours** (No KD)|79.8 $\pm$ 6.9|97.6 $\pm$ 2.3|70.9 $\pm$ 10.5|
> |**Ours** (FD)|81.9 $\pm$ 5.1|**99.1** $\pm$ 0.4|73.3 $\pm$ 7.5|
> |**Ours** (RD)|**84.4** $\pm$ 1.3|97.3 $\pm$ 1.7|**77.9** $\pm$ 1.8|
> |**Ours** (FD+RD)|83.3 $\pm$ 1.4|97.7 $\pm$ 1.0|76.1 $\pm$ 2.1|
>
> Even on categories to which CLIP has never seen, leveraging the text modality allow both SpectralGCD and GET to improve upon unimodal techniques. We added these results in **Table 15 (Appendix E.1)**.
>
> For these experiments we set the distillation weight $\lambda_{\text{kd}} = 0.5$ (instead of 1.0 as on other datasets). Because NEV includes categories not present in CLIP’s pretraining data, the teacher is less reliable and a strong distillation signal hinders student learning. Lowering the weight gives more reliable results while still avoiding semantic drift. We report the ablation of $\lambda_{\text{kd}}$ for CUB and Cars in **Table 6** of the revised paper. For the NEV dataset have added the corresponding ablation in **Table 16 (Appendix E.1)**.
>
> [1] Wang, Enguang, et al. "Get: Unlocking the multi-modal potential of clip for generalized category discovery." CVPR, 2025.
>
> ## On the definition of "New" accuracy (Q1)
>
> We follow the protocol introduced in the original GCD paper [2]. Clustering accuracy is computed by matching model predictions $\hat{y}_i$ to the ground-truth labels $y_i$:
>
> $$\text{ACC} = \max_{p \in \mathcal{P}(\mathcal{Y_u})} \frac{1}{M} \sum_{i=1}^{M} \mathbf{1} [ y_i = p(\hat{y}_i) ],$$
>
> where $\mathcal{D_u}$ is the unlabeled set, $M = |\mathcal{D_u}|$ and $\mathcal{P}(\mathcal{Y_u})$ is the set of all label permutations for the unlabeled classes $\mathcal{Y_u}$. Accuracy is then reported for three subsets: **All**: all instances in $\mathcal{D_u}$; **Old**: instances in $\mathcal{D_u}$ whose labels belong to the labeled set $\mathcal{Y_l}$; and **New**: instances in $\mathcal{D_u}$ whose labels belong to the novel classes $\mathcal{Y_u} \setminus \mathcal{Y_l}$. We added this definition in **Appendix A**.
>
> [2] Vaze, Sagar, et al. "Generalized category discovery." CVPR, 2022.

---

> > ### Comment · Reviewer_ei3h · 2025-11-25
> > **Rebuttal Acknowledgement**
> >
> > I have read the rebuttal. All my concerns have been addressed.

---

### Author Response · Authors · 2025-11-22
**Summary of changes**

We thank all reviewers for their comments and constructive criticism of our submission. Their invaluable input has helped us improve our work thanks to their insights and probing questions. In summary, on the basis of reviewer comments, we have:

+ enhanced motivations for why spectrally filtered cross-modal representations are discriminative with additional empirical evidence (**Main Paper Section 4.2** and **Appendix J**);
+ incorporated new experiments evaluating spectral filtering under partial splits (an incremental scenario) and under new/old class imbalance (**Appendix I** and **Appendix H**);
+ expanded computational analysis, including inference-time efficiency (**Appendix F**) and scalability in terms of number of concepts of our approach (**Appendix D**).
+ added additional ablations on student backbone capacity (**Main paper**, **Section 5.3**);
+ included additional analysis on the distillation weight (**Main paper**, **Section 5.3**);
+ extended dictionary evaluation, incorporating experiments using WordNet (**Appendix D**);
+ performed a fairness assessment of using a strong teacher through experiments on the NEV dataset unseen by CLIP (**Appendix E.1**); and
+ included new experiments jointly fine-tuning the text encoder together with the vision encoder (**Appendix G**).

**Note on References and Revisions:** All tables, figures, and sections cited throughout this rebuttal refer to the **revised manuscript**. To facilitate review, all additions and modifications to the PDF manuscript are highlighted in *blue* for easy identification.

---

### Author Response · Authors · 2025-12-01
**Summary of Author–Reviewer Discussion and Revisions**

We thank the Area Chairs and all reviewers for their effort and would like to take this opportunity to provide a summary of our interactions thus far. We believe the substantive engagement from three reviewers, along with the comprehensive experimental and expository additions to our manuscript, demonstrate the merits of our contribution.

### **Reviewer ei3h (Rating: 8)**: Fully satisfied with all responses.

Reviewer ei3h confirmed on November 25: *"All my concerns have been addressed."* Our additional experiments with generic, non-vision-centric dictionaries (WordNet) and evaluation on datasets unseen by CLIP (NEV) directly resolved their questions about generalizability and task-agnosticity.

### **Reviewer YwSE (Rating: 4):** Extensive technical discussion that substantially improved manuscript clarity.

Reviewer YwSE raised important technical questions about (1) the discriminativeness of Spectral Filtering when applied to our cross-modal representation; and (2) the appropriateness of applying CLIP for GCD given that classes should be "autonomously" discovered. These concerns prompted deeper reflections on our contributions:
- **On discriminativeness (1)**: We showed that our *cross-modal representations* achieve comparable separability to image features when used for linear probing, while demonstrating superior reliability in image-retrieval (6-14% mAP improvement). Additionally, an analysis of concept ranking by our Spectral Filtering method on WordNet showed that discriminative concepts are prioritized while background ones are filtered out.
- **On CLIP and autonomous discovery (2)**: We clarified that SpectralGCD never accesses new-class names during training, fully adhering to GCD assumptions. Even when the zero-shot teacher is given oracle access to all class names, SpectralGCD substantially outperforms the zero-shot baseline (+6.6% ImageNet-100, +19.8% Aircraft, +40% NEV). This highlights that effective multimodal GCD still requires discovering category structures from cross-modal cues and that it cannot simply rely on zero-shot performance, much like unimodal GCD methods have to refine strong pretrained priors without using true labels.

These discussions significantly enhanced the clarity and rigor of our manuscript, and we are grateful to Reviewer YwSE for prompting these reflections.

### **Reviewer GDAc (Rating: 4)**: Addressed majority of concerns, explicit score increase.

Reviewer GDAc raised questions related to (1) novelty and teacher capacity and (2) concerns about performance when ViT-B/16 is used as a teacher. We addressed Reviewer concerns as follows (note that *the Reviewer expressed their willingness to raise their rating to 6 on the basis of our response*):

- **On novelty and teacher capacity (1)**: Through extensive discussion, including new ablations on student backbone capacity, experiments on the NEV dataset, and detailed inference timing analysis, the reviewer stated on November 28: "I agree that some techniques are novel to the area... I will raise my score to 6 and leave further assessment to the other reviewers and the AC."
- **On performance with a ViT-B/16 teacher (2)**: The reviewer's remaining concern centers on performance when ViT-B/16 serves as both teacher and student. We note that this self-distillation configuration is an *intentionally out-of-design ablation* intended to isolate our Spectral Filtering and cross-modal representation contributions from teacher distillation. And we highlight that, even in this weakened setting, *SpectralGCD still outperforms SimGCD on two out of three datasets*, demonstrating that our gains derive from our methodological contributions rather than simply using a stronger teacher.



### **Reviewer hK7S (Rating: 4)**: No engagement in discussion as of November 27.

We provided a detailed response on November 22 addressing all weaknesses perceived by the Reviewer, including illustrative new experiments and tables, and thorough technical explanations. Reviewer hK7S did not respond to or acknowledge our rebuttal before the ability to do so was disabled.

We again thank the Area Chairs and all reviewers for their careful consideration during this uniquely challenging ICLR review period.

---

### Meta-Review · Area_Chair_5JxA · 2026-01-07

**Summary:**

This paper provides a promising method to inject external knowledge from VLM to improve GCD performance. After the rebuttal period, the paper received mixed scores of 8, 4, 6, and 4. Two of the reviewers did not participate in the rebuttal discussion, and the majority of their concerns were addressed in the authors’ responses. Taking this into account, the AC considers the paper to be borderline accepted.

**Reviewer Concerns:**

--addressed--： On performance with a ViT-B/16 teacher （Reviewer GDAc）

**Reviewer Scores:**

Regarding the comment from Reviewer hk75 concerning the sensitivity of SpectralGCD to imbalanced Old/New class ratios in the unlabeled data, the authors provided additional experimental results during the rebuttal that explicitly address this question. These results clarify the behavior of the method under significant distribution shifts, including scenarios where New classes dominate the unlabeled set. Had the reviewer been able to participate fully in the rebuttal discussion, these clarifications would likely have led to a revision of their initial score.

---

### Decision · Program_Chairs · 2026-01-26

Accept (Poster)